# Altered tRNA processing is linked to a distinct and unusual La protein in *Tetrahymena thermophila*

Kyra Kerkhofs [1], Jyoti Garg[1], Étienne Fafard-Couture[2], Sherif Abou Elela [3], Michelle S. Scott[2], Ronald E. Pearlman [1] & Mark A. Bayfield [1] ✉

Nascent pre-tRNAs are transcribed by RNA polymerase III and immediately bound by La proteins on the UUU-3'OH sequence, using a tandem arrangement of the La motif and an adjacent RNA recognition motif-1 (RRM1), resulting in protection from 3'-exonucleases and promotion of pre-tRNA folding. The *Tetrahymena thermophila* protein Mlp1 has been previously classified as a genuine La protein, despite the predicted absence of the RRM1. We find that Mlp1 functions as a La protein through binding of pre-tRNAs, and affects pre-tRNA processing in *Tetrahymena thermophila* and when expressed in fission yeast. However, unlike in other examined eukaryotes, depletion of Mlp1 results in 3'-trailer stabilization. The 3'-trailers in *Tetrahymena thermophila* are uniquely short relative to other examined eukaryotes, and 5'-leaders have evolved to disfavour pre-tRNA leader/trailer pairing. Our data indicate that this variant Mlp1 architecture is linked to an altered, novel mechanism of tRNA processing in *Tetrahymena thermophila*.

RNA polymerase III transcription of nascent pre-tRNAs terminates after the synthesis of a stretch of uridylates on a 3'-trailer extension (UUU-3'OH). La is the first protein to bind these nascent pre-tRNAs on the uridylate stretch[1] and assists with pre-tRNA folding[2–4] in a manner that can protect pre-tRNAs from 3'- to 5'-exonuclease digestion[5] and rescue defective pre-tRNAs from nuclear surveillance[6]. Once the nascent pre-tRNA has acquired a tRNA-like structure, the endonuclease RNase P removes the 5'-leader of La-bound pre-tRNA, followed by endonucleolytic cleavage of the 3'-trailer by RNase Z[7,8]. As a result of 3'-end cleavage, the La protein no longer associates with the tRNA and can be recycled for processing of new nascent pre-tRNAs[4]. In addition to the La-dependent pathway, an alternative La-independent pre-tRNA processing pathway exists but the order of pre-tRNA processing is reversed: without 3'-trailer protection by La, the exonuclease Rex1 digests the 3'-trailer of the pre-tRNA rapidly before RNase P endonucleolytic cleavage of the 5'-leader[5], and misfolded pre-tRNAs are also more prone to degradation through nuclear surveillance[6]. La binding to pre-tRNAs is therefore hypothesized to establish and determine the order of 5'-leader versus 3'-trailer processing[8]. Once the 5'-leader and 3'-trailer sequences are processed, tRNA nucleotidyltransferase adds a CCA sequence to the discriminator base at the 3'-end of the mature tRNA which will serve as the site of amino acid charging[9].

Genuine La proteins are members of the La-related proteins (LARPs)[10]. Most LARP family members, and all genuine La proteins, contain an N-terminal La module consisting of two adjacent RNA-binding domains: a La motif (LaM) and an RNA-recognition motif-1 (RRM1) (Fig. 1a)[11–13]. In *Tetrahymena thermophila*, the LARP7 ortholog p65 was until recently the only characterized LARP in this species[10,14], but a recent study has grouped the *T. thermophila* Macronucleus localized protein of unknown function (Mlp1)[15] with the genuine La proteins, based on its primary sequence conservation in the LaM[14]. Interestingly, Mlp1 only contains a highly conserved LaM, unlike all previously studied La proteins which contain the tandem LaM-RRM arrangement. This atypical La protein has been identified in several members of the alveolates[14]. Utilizing the interesting and important model organism, the alveolate ciliate protozoan *T. thermophila*

[1]Department of Biology, Faculty of Science, York University, Toronto, ON M3J 1P3, Canada. [2]Département de Biochimie et de Génomique Fonctionnelle, Faculté de Médecine et des Sciences de la Santé, Université de Sherbrooke, Sherbrooke, QC J1E 4K8, Canada. [3]Département de Microbiologie et d'Infectiologie, Faculté de Médecine et des Sciences de la Santé, Université de Sherbrooke, Sherbrooke, QC J1E 4K8, Canada. ✉e-mail: bayfield@yorku.ca

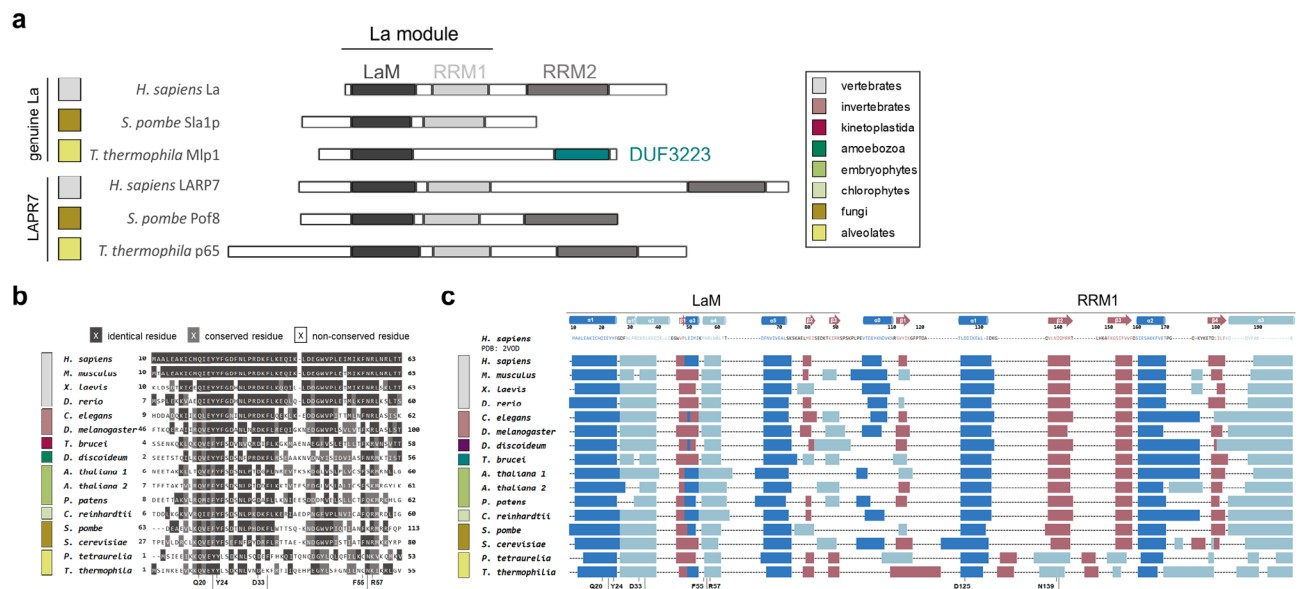

**Fig. 1 | Alveolate La proteins are predicted to have a LaM but not an RRM1.**
**a** Schematic representation of RNA-binding domains found in La and La-related protein-7 (LARP7) from different eukaryotes. The LaM and RRM1 together form the La module responsible for uridylate binding through formation of a hydrophobic binding pocket between the two domains. LaM La motif, RRM RNA-recognition motif, DUF domain of the unknown function. **b** Primary amino acid sequence alignments from different eukaryotic lineages showing conservation of uridylate binding residues (highlighted, bottom). A dark gray background indicates identical residues, light gray conserved residues, and white indicates no conservation. Color-coded legend is shown in (**a**). Full LaM and RRM1 domains are shown in Supplementary Fig. S1a, b. **c** Secondary structure predictions of LaM and RRM1 from different eukaryotic lineages. Predicted β-sheets shown in red and predicted α-helices in blue (dark blue: typical α-helices found in the winged-helix fold and classic RRM, light blue: inserted α-helices found in La proteins specifically) are compared against the secondary structure motif of the hLa protein crystal structure on the top (PDB: 2VOD). The location of the conserved amino acid residues important for uridylate binding are shown at the bottom. Color-coded legend is shown in (**a**).

(reviewed in ref. 16), the macronucleus/somatic nucleus localization and structural characteristics of Mlp1 could provide insight into the observation that despite the apparent lack of the RRM1, Mlp1 functions as a genuine La protein in tRNA processing in the nucleus.

The absence of the RRM1 in the La module of a genuine La protein in alveolates is highly unexpected, due to the mechanism by which La proteins bind the uridylate-containing 3'-trailer. Structural studies of the human La (hLa) LaM-RRM bound to UUU-3'OH revealed that both domains sandwich this RNA, and functional studies have demonstrated that the La module is indispensable and sufficient for uridylate binding[11,17], as deletion of either the LaM or RRM1 results in complete loss of binding[11,12,18,19]. Another unusual feature of this new type of La protein is the presence of a domain of unknown function-3223 (DUF3223) in the C-terminal region[14]. Thus, questions remain as to whether Mlp1 functions as a genuine La protein, and if so, whether it uses a mode of RNA binding dissimilar to previously studied La proteins, with possible associated changes in how Mlp1 directs pre-tRNA processing.

Here, we present evidence that despite the apparent lack of the RRM1, Mlp1 functions as a genuine La protein. Using ribonucleoprotein (RNP) immunoprecipitation (RIP)-Seq of Mlp1, we show an association with UUU-3'OH containing pre-tRNAs in vivo, and preferential binding of pre-tRNA substrates over mature tRNA substrates in vitro. In order to assess whether this altered architecture might have associated consequences in pre-tRNA processing, we tested Mlp1 function in a well-established model system and demonstrated that heterologous Mlp1 expression in *Schizosaccharomyces pombe* promotes pre-tRNA processing and tRNA-mediated suppression, but without typical La-associated 3'-end protection. Consistent with this, the genetic depletion of Mlp1 in *T. thermophila* reveals that in contrast to previously studied La proteins, Mlp1 destabilizes pre-tRNA 3'-ends, thus acting as a factor that accelerates 3'-end processing. In addition, when comparing pre-tRNAs found in *T. thermophila* with other eukaryotic species, we found that the 3'-trailer sequences are considerably shorter,

and that 5'-leader sequences have evolved accordingly to disfavor base pairing at these shortened 3'-trailers. Our data are consistent with a model in which Mlp1 fulfills many expected functions of a genuine La protein but uses alternate RNA-binding modes to promote a pre-tRNA processing pathway in *T. thermophila* that differs from those found in other studied eukaryotic systems.

## Results

### Mlp1 is predicted to lack an RRM adjacent to the LaM
A recent phylogenetic study in eukaryotes of all kingdoms identified Mlp1 as a new atypical genuine La protein in alveolates, containing a highly conserved LaM without an adjacent RRM[14]. To investigate primary amino acid sequence conservation, multiple sequence alignments were conducted for La proteins from different eukaryotic lineages. We confirm that residues important for uridylate binding in the LaM are mostly conserved in *T. thermophila* (Fig. 1b and Supplementary Fig. S1a, c, d). In contrast, we found little to no conservation of residues in the region of the adjacent domain where the RRM1 would be located (Supplementary Fig. S1b)[20]. When comparing primary sequences of the La module between different LARPs, the presence of an RRM1 domain could often only be inferred from secondary structure predictions[10]. Therefore, we compared the secondary structure predictions for *T. thermophila* La with secondary structures for different eukaryotes (Fig. 1c). These consistently predicted an RRM fold immediately C-terminal to the LaM in all species examined, except for candidate alveolate La proteins.

### Mlp1 preferentially binds pre-tRNAs in *T. thermophila*
We hypothesized that should Mlp1 function as a genuine La protein, it should bind and promote the processing of RNA polymerase III transcripts, as has been demonstrated in budding and fission yeast[7,8]. We immunoprecipitated Mlp1-associated RNAs followed by detection by northern blot (Fig. 2a and Supplementary Fig. S2a). Using probes specific for pre-tRNA 3'-extensions, we confirmed that Mlp1

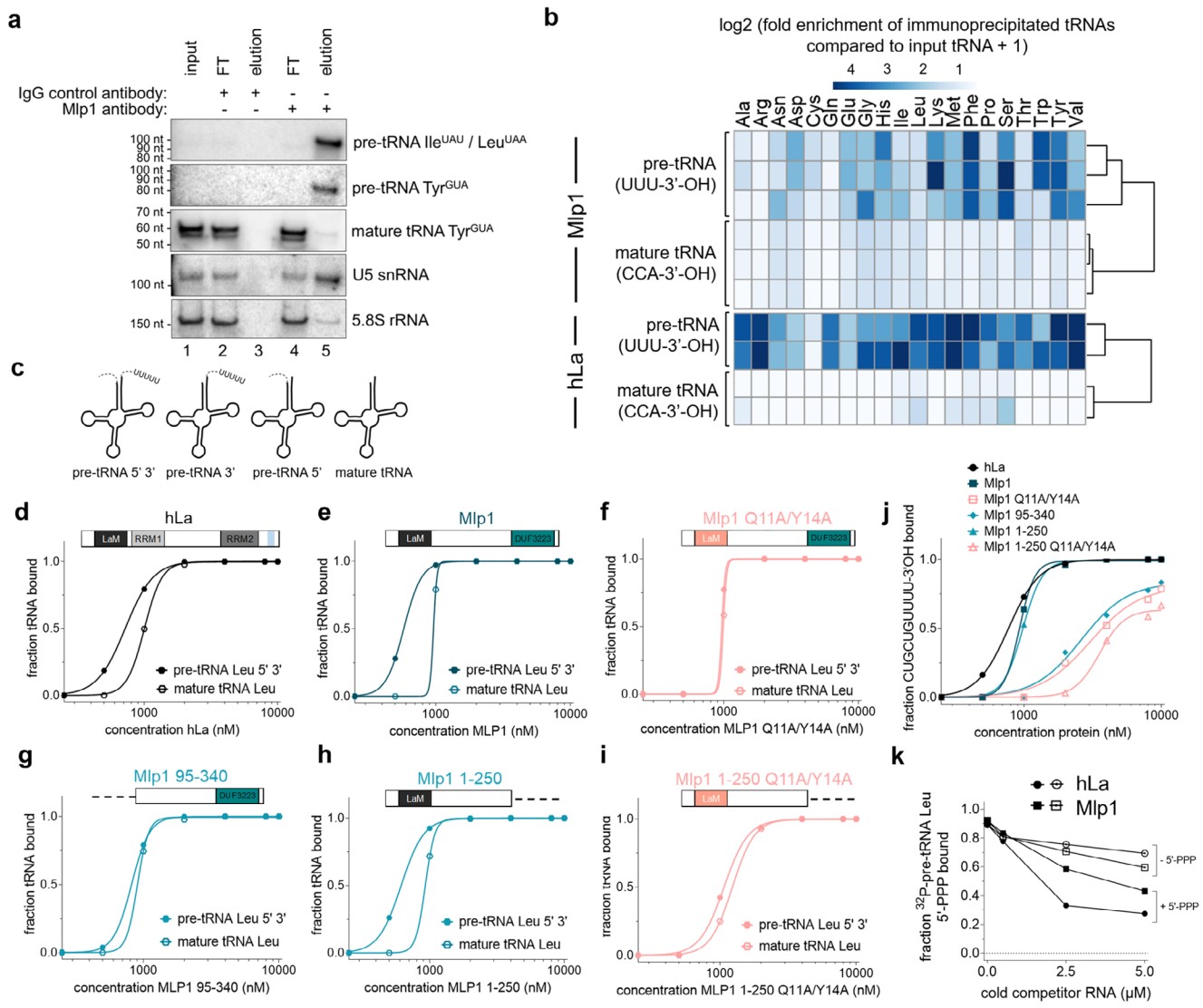

**Fig. 2 | Mlp1 preferentially binds pre-tRNAs partially via the 3′-terminal uri-dylates. a** Northern blot analysis of Mlp1 ribonucleoprotein-immunoprecipitated (RIP) samples from *T. thermophila*. Mlp1 enriches pre-tRNAs more efficiently than mature tRNAs ($n = 3$ biologically independent samples). Pre-tRNA Ile^UAU and Leu^UAA are both recognized by the 3′-trailer probe used. 5.8S rRNA was used as a non-binding loading control. FT flowthrough, IgG immunoglobin G control antibody. Western blot confirming Mlp1-specific immunoprecipitation is shown in Supplementary Fig. S2a. **b** Next-generation sequencing data of immunoprecipitated pre-tRNAs (rows represent biologically independent replicates), shown as a heatmap of log2 + 1 transformed fold enrichment calculated by taking ratios of normalized Mlp1-immunoprecipitated tRNA and input tRNA counts per million (CPM) for different tRNA isotypes (top). Next-generation tRNA sequencing data from ref. 22 analyzed similarly for hLa (bottom). The log2 + 1 transformed fold enrichment was calculated by taking ratios of normalized La-immunoprecipitated tRNA and the averaged input tRNAs CPM. The data were split between premature and mature tRNAs based on the 3′-end of the transcript (-CCA-ending mature tRNAs and

U-ending pre-tRNAs). Reproducibility between replicates for *T. thermophila* was determined by correlation plots shown in Supplementary Fig. S3. **c** Schematic representation of ^32P-labeled *T. thermophila* pre-tRNA intermediates containing 5′-leader and 3′-trailer, 5′-leader only, 3′-trailer only and mature tRNA used in **d**–**i**, **k**, Supplementary Fig. S2d, e, and Fig. 3a, b. **d**–**i** Binding curves from EMSAs comparing ^32P-labeled Leu^AAG 5′-leader, 3′-trailer-containing pre-tRNA and mature Leu^AAG tRNA ($n ≥ 2$ independent experiments). Native gels are shown in Supplementary Fig. S4a and Kd quantification in Table 1. **j** Binding curves comparing binding of hLa, Mlp1, and Mlp1 mutants to ^32P-labeled CUGCUGUUUU-3′OH RNA ($n ≥ 2$ independent experiments). Native gels are shown in Supplementary Fig. S4a and Kd quantification in Table 1. **k** Binding curves from competition EMSA between ^32P-labeled 5′-leader containing pre-tRNA containing a 5′-triphosphate (+5′PPP) and unlabeled competitors +5′PPP as a positive control and 5′-triphosphate removed pre-tRNA (−5′PPP) ($n = 2$ independent experiments). Native gels are shown in Supplementary Fig. S4d, e. Source data are provided as a Source Data file.

immunoprecipitation enriched pre-tRNA species relative to mature tRNAs, which were relatively more abundant in input RNA fractions. We also observed lesser enrichment of the U5 small nuclear RNA (snRNA), which is consistent with data indicating association of La proteins with processing intermediates of U5 in *Saccharomyces cerevisiae*[21]. We conclude that based on its expected RNA target cohort, Mlp1 behaves as expected for a genuine La protein.

To further investigate Mlp1-associated RNAs, we sequenced Mlp1-immunoprecipitated RNAs after removing RNAs >300 nt by

gel electrophoresis. As expected for a genuine La protein, we observed an enrichment of reads mapping to pre-tRNA genes carrying 3′-uridylate extensions relative to their normalized abundance in size-matched input samples (Fig. 2b−Mlp1). In contrast, mature tRNAs are not enriched in Mlp1-bound samples (Fig. 2b−Mlp1; individual values for tRNA isotypes are presented in Supplementary Fig. S2b). We compared our dataset with a previous study in which hLa-bound tRNA reads were obtained by photoactivatable cross-linking and immunoprecipitation (PAR-CLIP) and where tRNA

**Table 1 | K$_d$ values from EMSAs determining binding of hLa, Mlp1, and Mlp1 mutants to typical La protein–RNA targets**

| | Protein | CUGCUGUUUU-3'OH K$_d$ ± S.D. (nM) | pre-tRNA Leu K$_d$ ± S.D. (nM) | mature tRNA Leu K$_d$ ± S.D. (nM) | Relative binding K$_{rel (pre-tRNA vs mature tRNA)}$ |
|---|---|---|---|---|---|
| ■ | hLa | 772 ± 31 | 720 ± 44 | 1002 ± 16 | 1.39 |
| ■ | Mlp1 | 926 ± 27 | 586 ± 15 | 831 ± 33 | 1.42 |
| ■ | Mlp1 Q11A/Y14A | 3038 ± 416 | 965 ± 16 | 990 ± 15 | 1.03 |
| ■ | Mlp1 95-340 | 2616 ± 412 | 700 ± 62 | 902 ± 59 | 1.29 |
| ■ | Mlp1 1-250 | 982 ± 40 | 613 ± 53 | 927 ± 29 | 1.51 |
| ■ | Mlp1 1-250 Q11A/Y14A | 3554 ± 380 | 1068 ± 51 | 1232 ± 118 | 1.15 |
| ■ | Mlp1 1-95 | > 40,000 | N.D. | N.D. | N.A. |
| ■ | Mlp1 95-250 | > 40,000 | N.D. | N.D. | N.A. |

K$_d$ equilibrium dissociation constant, S.D. standard deviation, N.D. binding affinity not determined, N.A. relative binding calculation not applicable.
Color code corresponds to Fig. 2d–j.

sequencing of input and immunoprecipitated RNA was done using hydro-tRNAseq[22] and found a similar enrichment for 3'-uridylate-containing pre-tRNAs (Fig. 2b—hLa; individual values for tRNA isotypes presented in Supplementary Fig. S2c). The relative enrichment of pre-tRNAs over mature tRNAs, are strongly indicative of Mlp1 functioning as a genuine La protein in *T. thermophila*.

### Mlp1 preferentially binds pre-tRNA substrates in vitro
The hLa protein preferentially binds pre-tRNA substrates through engagement of multiple sites, including the pre-tRNA body, 3'-trailer, and 5'-leader[4,23]. To determine whether enrichment of Mlp1-associated pre-tRNAs relative to mature tRNAs correlated with changes in affinity for such ligands, we compared binding affinity of Mlp1 for in vitro transcribed mature tRNA versus pre-tRNA substrates using electrophoretic mobility shift assays (EMSAs), as well as versions of these that included either or both of 5'-leader or 3'-trailer extensions (Fig. 2c). We found that Mlp1 preferentially binds pre-tRNAs containing both 5'- and 3'-extensions, followed by pre-tRNA containing the 3'-trailer. Lowest binding affinities were consistently found for mature tRNAs, confirming the in vivo preferential binding of pre-tRNAs (Supplementary Fig. S2d, e and Supplementary Table S1).

To further investigate Mlp1 tRNA binding, we compared the affinity of full-length Mlp1 or Mlp1 mutants to hLa for radioactively labeled tRNA targets by EMSA (Fig. 2d–i, Supplementary Fig. S4a, and Table 1). To test the importance of 3'-uridylate binding, we compared Mlp1 to an Mlp1 mutant in which conserved amino acids predicted to recognize the UUU-3'OH motif were substituted (Q11A/Y14A)[12,19] (see Fig. 1b and Supplementary Fig. S1d, Q20/Y23 numbering in hLa), as well as a mutant in which the entire LaM was deleted (Mlp1 95–340). We found that the increased affinity for pre-tRNAs was lost in the Mlp1 Q11A/Y14A mutant (compare Fig. 2e, f) as well as for the LaM deletion mutant (compare Fig. 2e, g). In contrast, the C-terminal DUF3223 is not important to discriminate between pre-tRNA and mature tRNAs since the removal of this domain (Mlp1 1–250) maintains the binding affinity difference (Fig. 2h), while the Q11A/Y14A mutations in the context of the deleted DUF3223 (Mlp1 1–250 Q11A/Y14A) again resulted in decreased affinity for pre-tRNAs (Fig. 2i). To investigate UUU-3'OH binding more directly, we compared these Mlp1 mutants in protein−RNA-binding experiments using a 3'-trailer sequence CUGU-GUUUU-3'OH and found that the predicted uridylate binding residues located in the LaM (Q11A/Y14A) were required for binding (Fig. 2j, Supplementary Fig. S4a, and Table 1). Additionally, binding of 3'-uridylate RNA occurs with the same affinity for Mlp1 1–250 compared to full-length Mlp1 (Fig. 2j), whereas affinity for uridylate RNA is lost when both domains are used individually (Mlp1 1–95 and Mlp1 95–250) (Supplementary Fig. S4b). These data suggest that despite the apparent lack of the RRM1, Mlp1 functions in a similar manner as the hLa

protein in preferentially binding pre-tRNA substrates, and that predicted, conserved uridylate binding residues in the LaM promote higher affinity binding associated with the UUU-3'OH motif.

The hLa protein is known to also interact with the 5'-triphosphate containing end of the nascent pre-tRNA through a short basic motif located in the C-terminal part of the protein[23]. We tested the affinity of the 5'-triphosphate for Mlp1 by competition EMSA, using a 5'-leader containing, 3'-trailer processed in vitro transcribed radioactively labeled tRNA (+5'PPP) and unlabeled competitor tRNAs with the 5'-triphosphate removed after phosphatase treatment (−5'PPP; +/− 5'PPP demonstrated in Supplementary Fig. S4c). The unlabeled −5'PPP substrate competed poorly relative to an unlabeled +5'PPP substrate for the radioactive +5'PPP substrate on hLa (Fig. 2k—compare −5'PPP and +5'PPP hLa, Supplementary Fig. S4d), but the difference in competition between these RNAs on Mlp1 was smaller (Fig. 2k—compare −5'PPP and +5'PPP Mlp1, Supplementary Fig. S4e). These data are consistent with a lesser degree of 5'-leader discrimination for 5'PPP in Mlp1 relative to hLa.

### Mlp1 binding to 3'-trailer RNAs is different from hLa
To further study the distinct binding modes, we performed competition experiments between [32]P-labeled uridylate RNA (U10) and unlabeled competitor pre-tRNA and mature tRNA substrates. We used hLa as a control and found that, as expected from the previous work[4], pre-tRNAs were a stronger competitor for the uridylate binding pocket than mature tRNAs (Fig. 3a and Supplementary Fig. S5a). In contrast, uridylate RNA was competed off Mlp1 using only low amounts of either pre-tRNA or mature tRNA (Fig. 3b and Supplementary Fig. S5a), suggesting that binding of uridylates on Mlp1 is weaker compared to hLa.

Previous high-resolution structural characterization of hLa bound to UUU-3'OH established that the penultimate uridylate has the greatest importance for sequence-specific, high-affinity binding[18,19]. Notably, this uridylate is the only residue in the UUU-3'OH motif that contacts the RRM1 in hLa (Fig. 3c). Previous work established that hLa has no binding affinity for 10-mers C10, G10 or A10, however, 20-mer versions of G20 and A20 have a considerable binding affinity for hLa[24]. To compare the importance of the position of the penultimate uridylate (two nucleotides from the 3'-end: U$_{-2}$), we performed competition EMSAs for hLa and Mlp1 using the radioactively labeled UUU-3'OH containing RNA CUGCUGUUUU-3'OH RNA (hence referred to as wild-type 4U) and unlabeled RNAs carrying specific U to C variations to the terminal uridylates in this sequence, as previous work indicated the U to C substitution had the greatest effect on RNA substrate discrimination by hLa[19], and we confirmed complete lack of binding affinity of Mlp1 for C10 (Supplementary Fig. S5b). As expected, mutating the penultimate uridylate (U$_{-2}$) into a cytidylate (U$_{-2}$C – CUGCU-GUUCU-3'OH) resulted in this RNA functioning as a poor competitor with hLa, whereas changes at the most 3'-terminal position (U$_{-1}$C) and

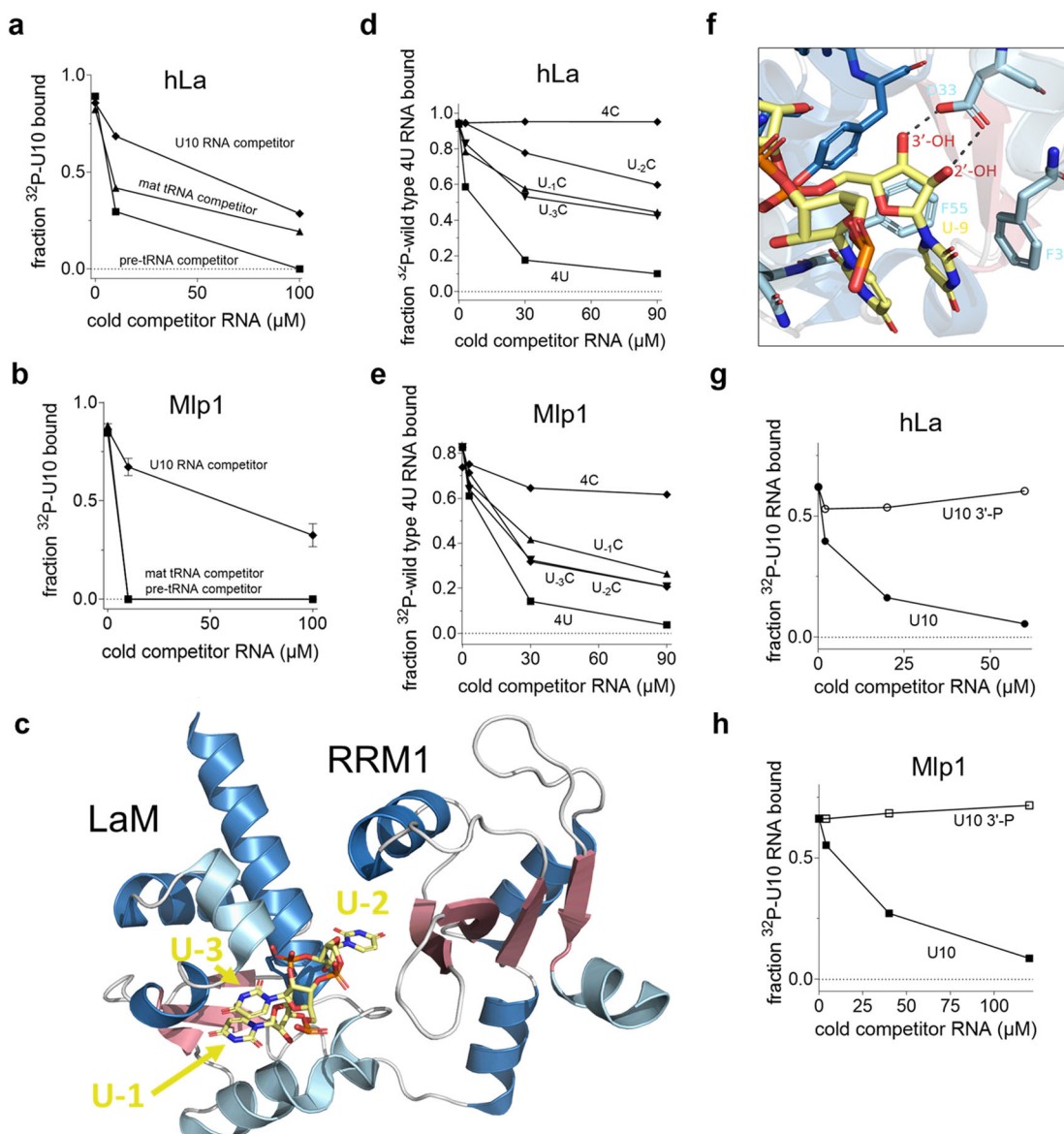

**Fig. 3 | Mlp1 does not discriminate 3'-uridylates as stringently as hLa.**
**a, b** Binding curves from competition EMSA between $^{32}$P-labeled uridylate RNA (U10) and unlabeled competitor U10 RNA (control), 5'-leader and 3'-trailer-containing *T. thermophila* pre-tRNA Leu$^{AAG}$ and mature tRNA Leu$^{AAG}$ for hLa (**a**) and Mlp1 (**b**). Error bars represent the mean +/− standard deviation (Mlp1 U10: $n = 4$, tRNA: $n = 2$, hLa U10: $n = 2$, tRNA: $n = 1$ independent experiments). Native gels are shown in Supplementary Fig. S5a. **c** Three-dimensional representation of the hLa protein in complex with uridylate RNA with the last three terminal nucleotides shown (UUU-3'OH) (PDB: 2VOD). The penultimate uridylate U$_{−2}$ is positioned in between the LaM and RRM1, while the 3'-terminal uridylate U$_{−1}$ is positioned more towards the outside in stacking formation with U$_{−3}$. RNA is shown in yellow; β-sheets are shown in red and α-helices shown in blue (dark blue: typical α-helices found in the winged-helix fold and classic RRM, light blue: inserted α-helices specifically found in La proteins). Image generated using PyMOL. **d, e** Binding curves from competition EMSAs using P$^{32}$-labeled wild-type CUGCUGUUUU (4U) and unlabeled 4U and mutant RNAs: CUGCUGUUUC (U$_{−1}$C), CUGCUGUUCU (U$_{−2}$C), CUGCUGUCUU (U$_{−3}$C) and CUGCUGCCCC (4 C) for hLa (**d**) or Mlp1 (**e**) ($n = 2$ independent experiments). Native gels are shown in Supplementary Fig. S5c, d. **f** Magnified views of hLa and 3'-terminal uridylate U$_{−1}$ interactions (PDB: 2VOD) demonstrating the importance of the 3'-end. Protein carbon backbones have the same color coding as in **c**, oxygen is shown in red, nitrogen is shown in blue, and phosphorous is shown in orange. RNA is shown in yellow. Image generated using PyMOL. **g, h** Binding curves from competition EMSA between P$^{32}$-labeled U10-3'OH (control) and U10-3'-P for hLa (**g**) and Mlp1 (**h**) ($n = 2$ independent experiments). Native gels are shown in Supplementary Fig. S5e. Source data are provided as a Source Data file.

the third last position (U$_{−3}$C) has a lesser effect on competition capability (Fig. 3d and Supplementary Fig. S5c). Mutating the four last positions (4 C – CUGCUGCCCC-3'OH) resulted in the inability to compete with the wild-type 4U RNA (Fig. 3d and Supplementary Fig. S5c). In contrast, competition for Mlp1 between radioactively labeled 4U and any unlabeled variant RNA competitor (U$_{−1}$C, U$_{−2}$C and U$_{−3}$C) showed similar competition levels, indicating that the position of the penultimate uridylate U$_{-2}$ is not as important for interactions between Mlp1 and 3'-trailer sequences (Fig. 3e and Supplementary

Fig. S5d). Interestingly, unlabeled competitor 4 C was more capable of competition with wild-type 4U RNA for binding on Mlp1. These data are also consistent with Mlp1 discrimination for uridylate RNA not occurring as strongly as for hLa, and that the specificity of binding to the penultimate uridylate is weaker or absent for Mlp1.

The previous high-resolution work on hLa in complex with UUU-3'OH containing RNA also revealed the importance of the 2'- and 3'-hydroxyls (-OH) for high-affinity binding in the hydrophobic binding pocket between LaM and RRM1[18,19], with two hydrogen bonds formed

with an aspartate (Fig. 3f) that is conserved in Mlp1 (see Fig. 1b and Supplementary Fig. S1d, D33 numbering in hLa). To investigate the importance of the free 3′-terminal hydroxyl groups for Mlp1 binding, we used competition EMSAs comparing a regular U10 and a 3′-phosphorylated unlabeled competitor RNA. We found that for both hLa and Mlp1 the phosphorylated substrate makes a poor competitor, confirming that the presence of a 3′-hydroxyl end is important for interactions between Mlp1 and uridylates (Fig. 3g, h and Supplementary Fig. S5e). Together, these results demonstrate that Mlp1 shows the same preference and LaM amino acid dependence for pre-tRNAs and UUU-3′OH binding as hLa, but that the altered architecture correlates with diminished discrimination for these substrates relative to hLa.

## RNA chaperone function of Mlp1 in an *S. pombe*-based heterologous system

Our computational and biochemical work suggested that Mlp1 differs from other examined La proteins in its binding and discrimination of RNA targets. We have previously used a well-established, heterologous tRNA-mediated suppression system in *S. pombe* to investigate the function of both yeast and human La in the promotion of La-dependent pre-tRNA processing. This system can report on La-dependent 3′-end protection of nascent pre-tRNAs from exonucleases, as well as RNA chaperone activity to enhance correct folding of nascent pre-tRNAs through the ability to rescue a misfolded suppressor tRNA in vivo[4,25–27]. To test whether the altered Mlp1 architecture correlated with differences in pre-tRNA processing, we transformed *S. pombe* La protein Sla1p, full-length Mlp1 and multiple Mlp1 mutants into a *sla1- S. pombe* strain (ySH9) which encodes a defective stop codon UGA-decoding suppressor tRNA (tRNA-Ser$^{UCA}$) as well as the *ade6-704* allele, which is decoded by a tRNA-Ser$^{UCA}$ to suppress red pigment accumulation during growth on low adenine. Successful suppression in this system relies on the presence of a La protein or other suitable RNA chaperone, resulting in white colonies versus red colonies in unsuppressed cells.

When comparing Sla1p-transformants to full-length Mlp1-transformants, we found that Mlp1 can stabilize the defective suppressor pre-tRNA similar to Sla1p (Fig. 4a). Maturation of the defective suppressor tRNA can occur via 3′-terminal protection of uridylates from exonucleases, or general RNA chaperone activity, or a combination of these to assist with folding of pre-tRNAs[6]. To test the importance of the uridylate binding residues for suppression, we compared the Mlp1 Q11A/Y14A mutant and the LaM deletion mutant (Mlp1 95–340) to wild-type Mlp1 and observed a pink phenotype indicating intermediate suppression levels despite equal levels of protein expression (Fig. 4a and Supplementary Fig. S6a), suggesting that these uridylate binding residues function in maturation of the pre-tRNA in vivo. Next, we studied the function of the C-terminal DUF3223 and found that removal of this domain (Mlp1 1–250) led to near complete loss of suppression (Fig. 4a), and combination of DUF3223 removal and uridylate binding inactivation (Mlp1 1–250 Q11A/Y14A) resulted in a complete loss of suppression (Fig. 4a). These results indicate that to obtain complete tRNA-mediated suppression, both the conserved uridylate binding residues (Q11 and Y14) in the LaM and the C-terminal DUF3223 are required.

To investigate 3′-end protection more directly, we extracted total RNA from Mlp1- and Sla1p-expressing strains and analyzed suppressor pre-tRNA-Ser$^{UCA}$ processing by northern blot (Fig. 4b). We found that higher levels of suppressor pre-tRNA stabilization in Sla1p- and Mlp1-transformed strains (lanes 2, 3) corresponded to more mature suppressor tRNA which correlates with the white phenotype observed in the tRNA-mediated suppression results (see Fig. 4a), as opposed to the lack of pre-tRNA stabilization and subsequent mature suppressor tRNA in pRep4 control (lane 1) resulting in a red phenotype (Fig. 4a, b). Interestingly, the most abundant pre-tRNA species was slightly smaller in Mlp1-transformed cells relative to Sla1p (see asterisk). We also

detected endogenous pre-tRNAs Lys$^{CUU}$ processing intermediates using an intron, 5′-leader, and a 3′-trailer probe (Fig. 4c). The presence of the three La-dependent pathway pre-tRNA intermediates are detected after transformation of the positive control Sla1p, as have been described previously[28] (Fig. 4c−intron probe, compare lanes 1 and 2, indicated by a, b, and c). Notably, we observed the appearance of a distinct pre-tRNA intermediate in Sla1p expressing cells when using a probe specific for the 5′-leader of pre-tRNA Lys$^{CUU}$ that did not co-migrate with any of the major species observed using the intron or 3′-trailer probe (Fig. 4c− 5′-leader, indicated by d, beneath the full-length pre-tRNA band). We hypothesize that this species represents a subset of pre-tRNAs that are not stably bound by Sla1p and whose 3′-ends have been nibbled by a 3′-exonuclease, yet have retained their 5′-leaders as has been described during La-independent pre-tRNA processing. As expected from the tRNA-mediated suppression assay, Mlp1 also stabilized endogenous precursor pre-tRNA Lys$^{CUU}$, however, this precursor tRNA co-migrated with the 5′-leader containing, 3′-trailer exonuclease processed intermediate observed in Sla1p-transformants (Fig. 4c−5′-leader, band d, compare lane 2 and 3), consistent with Mlp1 stabilizing a 5′-leader containing, 3′-processed or nibbled pre-tRNA intermediate. As expected, all Mlp1 mutants defective in tRNA-mediation suppression were defective in stabilizing pre-tRNA intermediates. Together, these data are consistent with Mlp1 engaging pre-tRNAs and promoting tRNA-mediated suppression in *S. pombe*, but with impaired protections of pre-tRNA 3′-ends relative to Sla1p.

To further compare Mlp1 and Sla1p function in the processing of pre-tRNA intermediates by Mlp1 in *S. pombe*, we immunoprecipitated Sla1p and Mlp1 RNP-complexes (Supplementary Fig. S6b) and sequenced the 3′-ends of their associated pre-tRNA Lys$^{CUU}$ and pre-tRNA Tyr$^{GUA}$ by 3′-rapid amplification of cDNA ends (3′-RACE). While Sla1p-associated pre-tRNAs were enriched for species containing primarily four and five-nucleotide long uridylate 3′-trailers, as has been described previously[29], Mlp1-associated pre-tRNAs were largely depleted for 3′-trailer-containing species (Fig. 4d), consistent with the northern blotting results (Fig. 4b, c), with a minority of Mlp1-immunoprecipitated pre-tRNAs ending in 3′-CCA, indicating Mlp1 can also bind pre-tRNAs after CCA addition but prior to nuclear export and intron removal. These data are consistent with our Mlp1 RNA-binding data in vitro, indicating that Mlp1 promotes pre-tRNA processing but with impaired 3′ discrimination and lower protection of the 3′-ends of associated pre-tRNAs relative to other examined genuine La proteins.

## Mlp1 depletion in *T. thermophila* leads to impaired 3′-trailer processing

To investigate the effect of Mlp1 depletion on pre-tRNA processing in *T. thermophila*, we generated a partial *MLP1* knockout strain and confirmed genomic integration of the selection marker by southern blot and PCR (Supplementary Fig. S7a, b and data not shown) and reduced protein expression of Mlp1 by western blot (Fig. 5a) and indirect immunofluorescent staining (Supplementary Fig. S7c). Complete deletion of the *MLP1* locus in *T. thermophila* macronuclei through increasing drug selection (phenotypic assortment[30]) was not achieved, indicating that Mlp1 is likely essential in this system, similar to *Mus musculus*, *Drosophila melanogaster*, *Trypanosoma brucei* and *Arabidopsis thaliana*[31–34]. We extracted total RNA followed by pre-tRNA intermediate detection by northern blot for pre-tRNA Ile$^{UAU}$, Leu$^{UAA}$, and Val$^{CAC}$. In the absence of Mlp1, a 3′-trailer-containing pre-tRNA intermediate was stabilized (Fig. 5b−intron probe, bottom band). These data provide evidence for *T. thermophila* being the first eukaryote in which La protein levels correlate inversely with 3′-trailer stabilization. These results are also consistent with our heterologous expression data in *S. pombe*, indicating impaired protection of pre-tRNA 3′-trailers by Mlp1. These data suggest that the variant domain architecture of Mlp1 relative to other genuine La proteins is associated with an altered pre-tRNA processing pathway in this system.

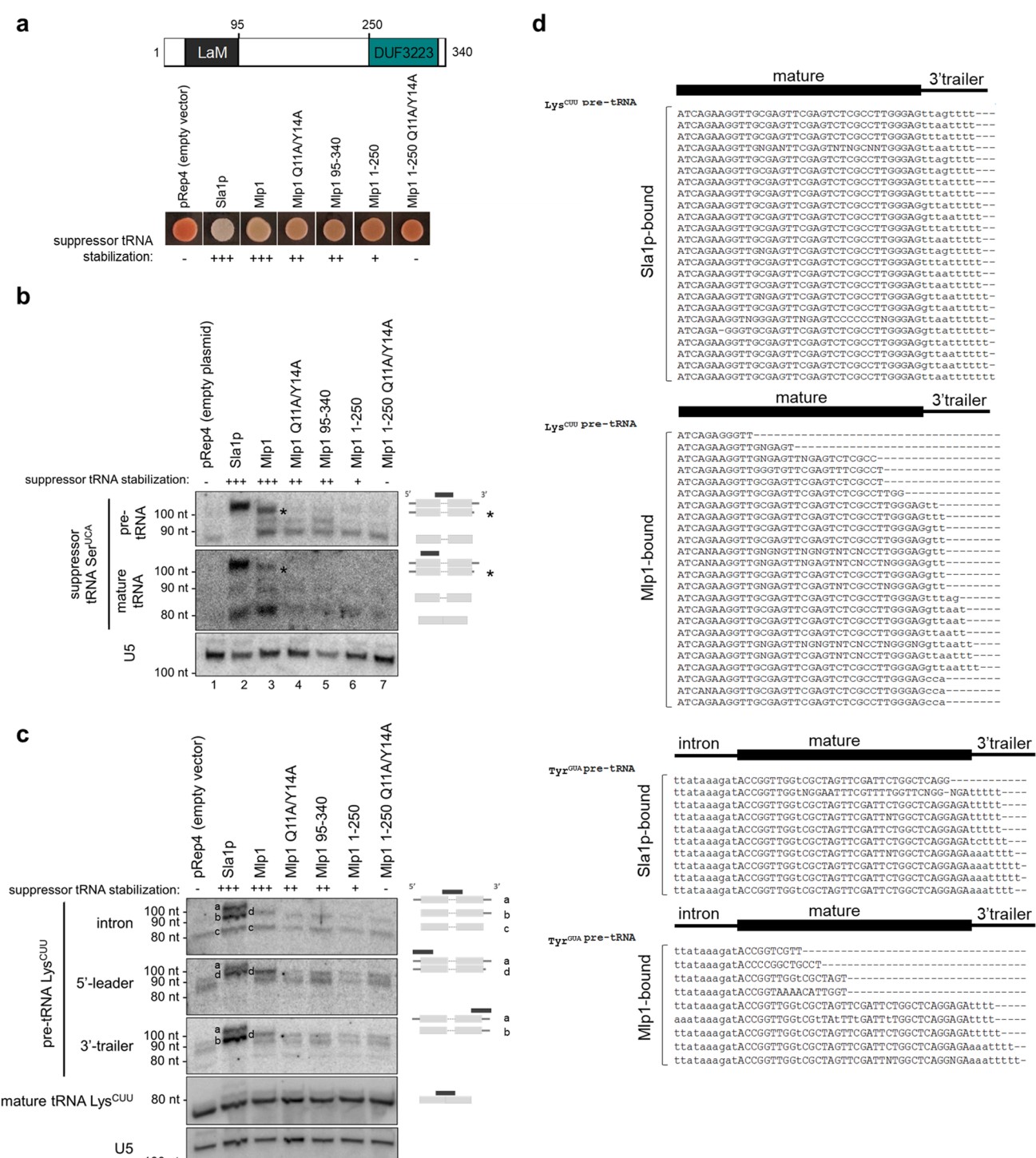

To study the effect of Mlp1 depletion on a transcriptome-wide scale, we performed total RNA-sequencing of small RNA-enriched samples, similar to our Mlp1 RIP-Seq (see Fig. 2b), in which counts for each pre-tRNA isoacceptor were determined for each unique uridylate tail length at the 3'-end of the tRNA. We found that the average length of the 3'-uridylate tail increased for the majority of tRNA species as a result of Mlp1 depletion (Fig. 5c, d). These findings indicate that Mlp1 promotes the removal of the 3'-trailer and that when Mlp1 is limiting, the 3'-trailer sequences are not processed as efficiently. Together, these data are consistent with Mlp1 accelerating 3'-end processing, relative to other examined species in which genuine La proteins stabilize 3'-trailers.

Previous work has demonstrated that in eukaryotes, the number of genomic tRNA gene copies correlates strongly with the expression of mature tRNA transcripts and thus can function as a predictor for tRNA expression levels[35,36]. We observed that in *T. thermophila* the number of genomic tRNA genes correlates with the expression levels of tRNAs ($R^2 = 0.55$, $r = 0.75$) (Fig. 5e). To investigate if depletion of Mlp1 results in changes of total tRNA levels, we performed the same analysis for our Mlp1 partial knockout strain ($R^2 = 0.53$, $r = 0.73$) (Fig. 5f) and found that when comparing tRNA expression between wild type and partial Mlp1 knockout strains the correlation was very strong ($R^2 = 0.94$, $r = 0.97$) (Fig. 5g), indicating that Mlp1 depletion did not influence mature tRNA expression. Northern blot analysis for

**Fig. 4 | Mlp1 promotes tRNA-mediated suppression in the absence of protection of 3′-trailers. a** tRNA-mediated suppression assay of *Schizosaccharomyces pombe* ySH9 strain transformed with pRep4 encoded Sla1p (positive control; *S. pombe* La protein), Mlp1 or indicated Mlp1 mutants (*n* = 3 biologically independent samples). Protein expression levels are shown by western blot in Supplementary Fig. S6a. Levels of suppressor tRNA stabilization is based on the color of the colonies and annotated as follows: full suppression (+++), near full suppression (++), intermediate suppression (+), no suppression (−). Top: diagram showing domain architecture of Mlp1. **b** Northern blot to determine 3′-end protection of suppressor pre-tRNA-Ser^UCA in ySH9 transformants shown in (**a**) (*n* = 2 biologically independent samples). Accumulation of suppressor pre-tRNA-Ser^UCA was determined using an intron complementary probe and mature tRNA using a tRNA body probe leading to the detection of both pre-tRNA and mature tRNA. The asterisk indicates a 3′-processed pre-tRNA intermediate. An excess unlabeled probe complementary to endogenous Ser^UGA was added to avoid cross-reaction between suppressor tRNA

and endogenous tRNA. U5: loading control. **c** Northern blot to determine 3′-end protection of endogenous pre-tRNA Lys^CUU in ySH9 transformants shown in (**a**), with suppressor tRNA stabilization levels (see panel **a**) shown above the blot (*n* = 3 biologically independent samples). Accumulation of pre-tRNA Lys^CUU intermediates was determined using an intron complementary probe which detects unprocessed 5′-leader and 3′-trailer-containing pre-tRNA (band a), 5′-leader processed pre-tRNA (band b) and both 5′-leader and 3′-trailer processed pre-tRNA (band c) in Sla1p transformants (positive control; lane 2). The 3′-processed, 5′-leader containing pre-tRNA intermediate is annotated as band d. The same blot was probed with a 3′-trailer complementary probe, a 5′-leader complementary probe and a mature tRNA complementary probe. U5: loading control. **d** Sanger sequencing of clonal isolates corresponding to cDNAs derived from 3′-terminal sequences from Sla1p- and Mlp1-immunoprecipitated pre-tRNAs in ySH9 as transformed in (**a–c**) (*n* = 2 biologically independent samples). Western blot of Sla1p- and Mlp1-immunoprecipitations shown in Supplementary Fig. S6b. Source data are provided as a Source Data file.

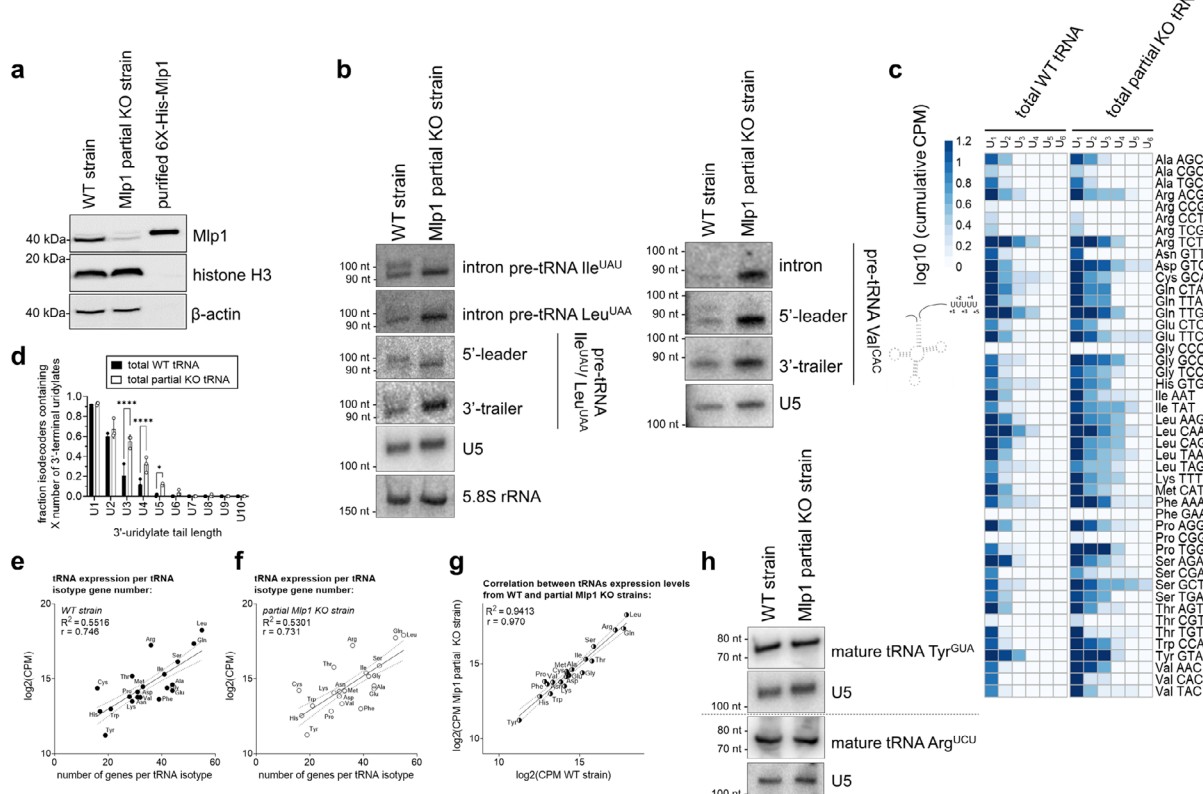

**Fig. 5 | Depletion of Mlp1 results in altered pre-tRNA processing in *T. thermophila* with normal levels of mature tRNAs. a** Western blot confirming decreased protein expression of Mlp1 in the partial Mlp1 knockout (KO) strain compared to the wild-type (WT) strain (*n* = 3 biologically independent samples). Loading control: histone H3 and β-actin. Purified 6X-His-Mlp1 was analyzed to control for antibody specificity. **b** Northern blot detecting tRNAs Ile^UAU, Leu^UAA, and Val^CAC pre-tRNA intermediates with an intron, 5′-leader, and 3′-trailer specific probe (*n* = 3 biologically independent samples). Pre-tRNA Ile^UAU and Leu^UAA are both recognized by 3′-trailer and 5′-leader probe used. U5 snRNA and/or 5.8 S rRNA were used as loading controls. **c** Next-generation sequencing data shown as a heatmap of log10 transformed normalized cumulative pre-tRNA counts for each uridylate tail length (*n* = 3 biologically independent replicates). For clarity, the average is plotted and the longest uridylate tail shown is U₆ (UUUUUU-3′OH). **d** Quantification of uridylate tail length from next-generation sequencing data (*n* = 3 biologically independent samples) in **c**. Error bars represent the mean +/− standard deviation. A two-way

ANOVA with Bonferroni's correction for multiple comparisons was performed to determine statistical significance: ****$P$ < 0.0001, *$P$ = 0.0491. For each tRNA isodecoder, the presence of a uridylate tail length (U1, U2, etc.) was scored as present if its abundance in CPM was greater than 0.3; log10(2). **e, f** Scatterplots of log2 transformed normalized tRNA counts (CPM) from next-generation sequencing data from WT strains (**e**) and partial Mlp1 KO strains (**f**) plotted against the number of genes per tRNA isotype encoded in the genome (see Supplementary Data 1). The correlation was calculated, and a positive correlation was observed for both WT and KO strains ($r$ = 0.746 and $r$ = 0.731, respectively) indicating that more tRNA gene copies result in higher tRNA expression. **g** Plotting of log2 transformed normalized WT and partial Mlp1 KO counts (CPM) gives a strong positive correlation ($r$ = 0.970). **h** Northern blot analysis detecting mature tRNA Tyr^GUA and Arg^UCU levels using a probe specific for the spliced mature tRNA (*n* = 3 biologically independent samples). U5 was used as a loading control on both membranes. Source data are provided as a Source Data file.

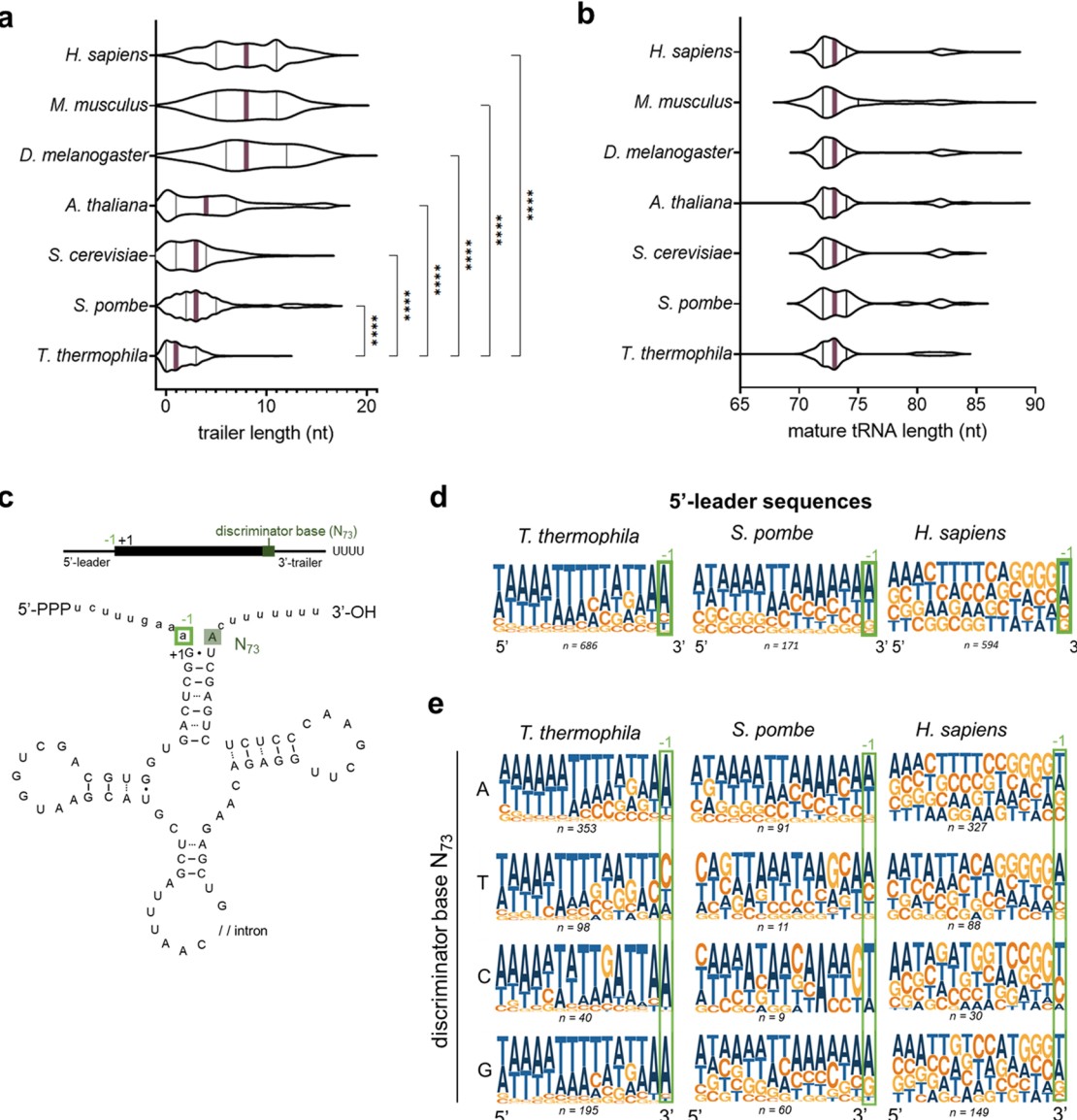

**Fig. 6 | The 3'-trailer length of _T. thermophila_ tRNAs is considerably shorter compared to other eukaryotes and affects the N$_{-1}$ composition of the 5'-leader.** **a** Genome-wide analysis of 3'-trailer lengths as determined by the number of nucleotides between the discriminator base (the most 3'-terminal nucleotide in the mature tRNA upstream of the posttranscriptional added CCA) and a genomic stretch of four Ts of different eukaryotes. Violin plots shows the median as a full purple line and the quartiles as full black lines. A one-way ANOVA with Tukey's correction for multiple comparisons was performed to determine statistical significance: ****$P < 0.0001$. Data summary shown in Table 2. **b** Genome-wide analysis of mature tRNA length for different eukaryotes. Violin plots show the median as a full purple line and the quartiles as full black lines. A one-way ANOVA with Tukey's correction for multiple comparisons was performed to determine statistical significance: no statistical significance ($P < 0.05$) was observed. Data summary is shown in Table 2. **c** Schematic representation of a pre-tRNA. The mature tRNA sequence is shown as a black rectangle and written in uppercase (UCGA) in the tRNA cartoon. Pre-tRNA-specific sequences, including the 5'-leader and 3'-trailer, are shown as a black line and written in lowercase (ucga) in the tRNA cartoon. The discriminator base (N$_{73}$) is the most 3'-terminal nucleotide of the mature tRNA sequence highlighted in dark green. The most 3'-terminal nucleotide of the 5'-leader sequence is highlighted with a light green box (N$_{-1}$). **d, e** Logo analysis of 5'-leader sequences of pre-tRNAs in _T. thermophila, S. pombe,_ and _H. sapiens_ (**d**). The same analysis split by the discriminator base identity (**e**). The number of pre-tRNAs representing each condition is shown underneath each logo. Source data are provided as a Source Data file.

representative tRNAs confirmed similar amounts of mature tRNA between wild-type and partial Mlp1 knockout strains (Fig. 5h), reminiscent of minimal differences in mature tRNA abundance (despite changes to pre-tRNA processing) in conditional mouse La knockout cells[37].

## The 3'-trailer lengths are shorter in _T. thermophila_ relative to other eukaryotes

To explore tRNAs and pre-tRNA processing in _T. thermophila_ more extensively, we compared their predicted pre-tRNA architecture to those from other eukaryotic species. We compared 3'-trailer lengths of

each pre-tRNA, as determined by the number of nucleotides between the discriminator base and the genomic stretch of at least four consecutive thymines[38] that give rise to the 3'UUU-OH motif in the nascent transcript. We found that _T. thermophila_ has shorter 3'-trailer lengths than any other species analyzed, with the most common 3'-trailer length being zero nucleotides (Fig. 6a, Table 2, and Supplementary Data 2). Identical trends were observed for genomic stretches/candidate terminators of five, six or seven consecutive thymines (Supplementary Table S2 and Supplementary Data 2). A similar analysis for the total length of the mature tRNA revealed that, as expected, the average length of mature tRNAs is the same as other examined species (Fig. 6b

**Table 2 | Summary of 3′-trailer and mature tRNA sequence lengths in different eukaryotic species**

| Species | Average of mature tRNA length ± S.D. (nt) | Number of tRNAs analyzed | Average of 3′-trailer length ± S.D. (nt) | Number of tRNAs analyzed (3′-trailer lengths with >20 nt excluded) |
|---|---|---|---|---|
| *H. sapiens* | 74 ± 3.4 | 594 | 8.1 ± 3.7 | 412 |
| *M. musculus* | 74 ± 3.7 | 811 | 7.9 ± 3.6 | 343 |
| *D. melanogaster* | 74 ± 3.6 | 295 | 8.5 ± 4.0 | 247 |
| *A. thaliana* | 74 ± 3.8 | 642 | 4.6 ± 4.3 | 532 |
| *S. cerevisiae* | 74 ± 3.5 | 275 | 3.0 ± 2.6 | 269 |
| *S. pombe* | 74 ± 3.4 | 171 | 3.3 ± 2.8 | 162 |
| *T. thermophila* | 74 ± 3.3 | 686 | 1.5 ± 1.5 | 686 |

*S.D.* standard deviation, *nt* nucleotides.

and Table 2). We previously noted differences in the enrichment of pre-tRNAs isotypes (Supplementary Fig. S2b−pre-tRNAs) in our Mlp1 RIP-seq results and hypothesized that the 3′-trailer length could be a determinant for binding affinity. We plotted fold enrichment for pre-tRNAs between Mlp1-immunoprecipitated and input tRNAs against 3′-trailer length and found that these are moderately negatively correlated ($R^2 = 0.14$, $r = −0.47$) (Supplementary Fig. S8a), suggesting that nascent pre-tRNAs containing a longer 3′-trailer sequence may have a slightly lower binding affinity for Mlp1. When comparing this to the hLa data, we found a lesser correlation ($R^2 = 0.05$, $r = −0.23$) (Supplementary Fig. S8b).

Processing of the 5′-leader by RNase P is based on tertiary structure recognition of the tRNA followed by endonucleolytic cleavage at the −1/+1 position[39,40]. Previous studies demonstrated structural conservation of the catalytically active RNA in RNase P in *T. thermophila*, indicating that processing occurs in a similar manner[40,41]. Optimal efficiency of RNase P cleavage is dependent on the presence of a bulge that includes the last nucleotide of the 5′-leader (the $N_{−1}$ position; Fig. 6c), as extensive base pairing between 5′-leader and 3′-trailer sequences inhibits 5′-leader cleavage[42]. We hypothesized that *T. thermophila* tRNAs should have adapted 5′-leader sequences to ensure the generation of a mismatched bulge against the shorter 3′-trailer sequences. We determined the most common 5′-leader sequences by logo generation for *T. thermophila, S. pombe,* and *Homo sapiens* and found that the most frequent nucleotide at the most 3′-residue of the 5′-leader sequence ($N_{−1}$) in *T. thermophila* is an adenine (±75%) (Fig. 6d and Supplementary Fig. S8c). In contrast, the distribution of nucleotides at this position is more diverse in other species (Fig. 6d and Supplementary Fig. S8c). At the 3′-end of the mature transcript, the discriminator base ($N_{73}$) is mostly adenine (±50%), followed by guanine as second most frequent nucleotide (±25%) (Supplementary Fig. S8d) which is common for all eukaryotes in this study.

We then split our data based on the identity of the discriminator base ($N_{73}$) (Fig. 6e and Supplementary Fig. S8e), which can pair with the first nucleotide in the 5′-leader ($N_{−1}$). We observed a strong lack of Watson−Crick base pairing for *T. thermophila* between the opposing nucleotides in the discriminator base pairing with the typical adenine ($A_{73} − A_{−1}$; $C_{73} − A_{−1}$, and $G_{73} − A_{−1}$). Conversely, tRNAs containing a thymine as discriminator base ($T_{73}$) strongly avoided an adenine as the $N_{−1}$ 5′-leader base to avoid base pairing. This partitioned discrimination is not evident for *S. pombe* and *H. sapiens* (Fig. 6e). Since shortened 3′-trailers in *T. thermophila* should have a greater dependence on leader-trailer bulges occurring through the discriminator base, these data are thus consistent with the 3′ most nucleotide of the 5′-leader having a greater evolutionary pressure to avoid base pairing with the discriminator base which is seen most strongly for *T. thermophila* (Supplementary Fig. S8f) relative to other examined species.

## Discussion

Previously studied genuine La proteins contain a conserved tandem arrangement of a LaM and RRM1 collectively referred to as a La

module. Previous phylogenetic predictions[14] and our continued computational analysis have indicated that the previously characterized *T. thermophila* protein Mlp1 may group with the genuine La proteins, despite the predicted lack of the La module's RRM1 domain. In addition to the absence of the RRM1, Mlp1 is predicted to have a previously uncharacterized domain of unknown function (DUF3223), which is absent in all other examined members of the LARP superfamily. Since this arrangement of a genuine La protein is unprecedented, it was not clear how this protein might perform La-associated functions and what effects this might have, if any, on the processing of La RNA targets.

We present data consistent with Mlp1 functioning as a genuine La protein. Using in vitro binding assays, we demonstrate that the identity of the Mlp1 residues Q11 and Y14, analogous to UUU-3′OH binding residues in hLa, are also required for high-affinity binding of 3′-uridylate-containing trailer sequences. We demonstrate that binding of uridylate RNA is retained when both the predicted LaM (Mlp1 1–95) and middle domain (Mlp1 95–250) are included, however, uridylate binding does not occur when either the LaM (Mlp1 1–95) or the middle domain (Mlp1 95–250) are tested in isolation. Given the relative paucity of contacts from the RRM relative to the LaM during hLa binding to UUU-3′OH, it is likely that the 95–250 region might serve an analogous function, making contacts important for UUU-3′OH binding despite the absence of the RRM fold.

While the Mlp1 LaM and middle domain combine to support UUU-3′OH binding similar to the classic La module, other key differences exist. Competition experiments indicated that short UUU-3′OH containing trailers were more easily displaced from Mlp1 by pre-tRNA and mature tRNA substrates, relative to hLa, suggesting that Mlp1 binding to UUU-3′OH may be less stable compared to La proteins with the classic LaM-RRM1 arrangement. The previous hLa-UUU-3′OH co-crystals revealed that RRM1 makes only a single contact with the penultimate uridylate RNA ($U_{−2}$) during UUU-3′OH binding, and it is this uridylate that is recognized with the greatest specificity. Unlike hLa, the $U_{−2}$C RNA was no less effective a competitor against the 4U RNA than the $U_{−3}$C and $U_{−1}$C RNAs when Mlp1 was tested. The impaired ability of Mlp1 to discriminate the $U_{−2}$ residue, as well as uridylates more generally, suggests that the lack of the RRM1 results in a relatively lower ability of Mlp1 to differentiate UUU-3′OH-containing RNAs.

Previous work has demonstrated that the La module (LaM + RRM1) of different LARPs, and more specifically their RRM regions, possess RNA chaperone activity in the absence of 3′-uridylate protection from exonucleases[25,26]. Using tRNA-mediated suppression in a La null strain (*sla1-*) of *S. pombe*, we demonstrated that while the uridylate binding residues in the LaM of Mlp1 are important for suppression, the DUF3223 region also promotes the correct folding of defective pre-tRNAs. These data raise the possibility that the DUF3223 region of Mlp1 may serve an analogous function in RNA chaperone activity previously associated with the RRM1 and other C-terminal regions of genuine La proteins.

Genuine La proteins are, however, well known to also stabilize 3′-trailer-containing intermediates through high-affinity binding of the 3′-

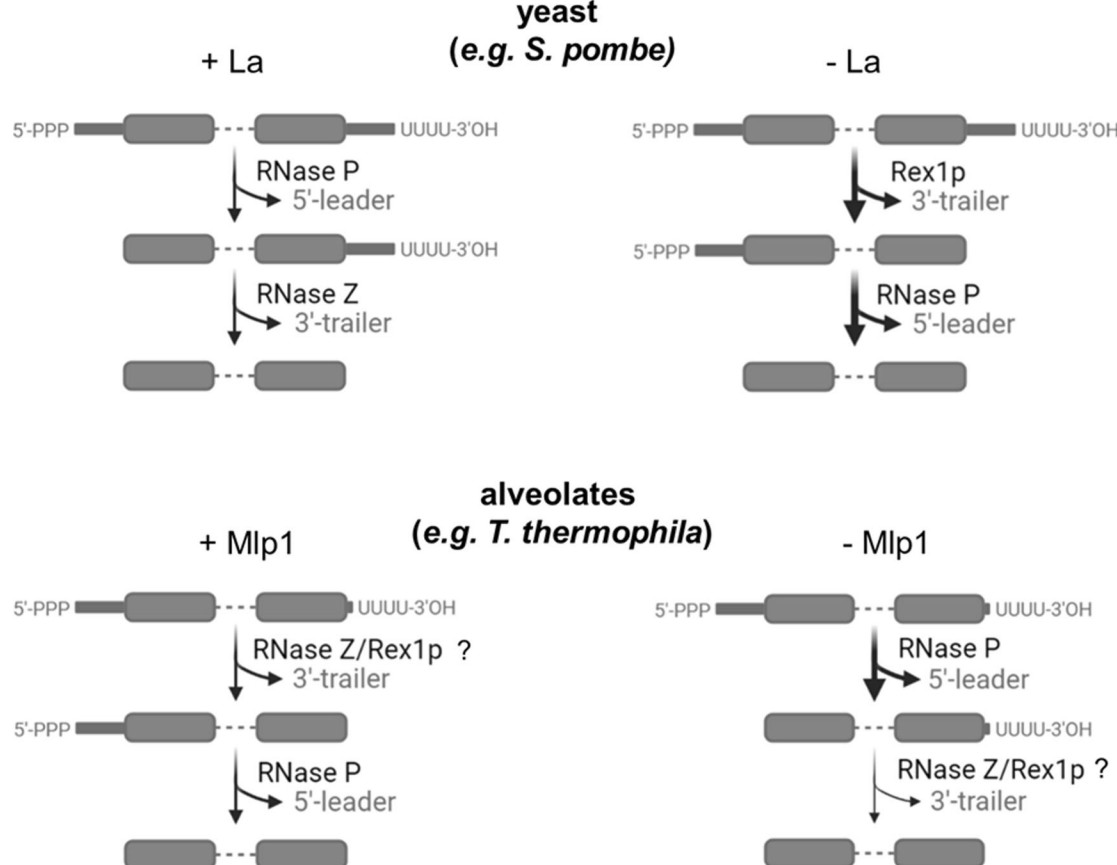

**Fig. 7 | Model for pre-tRNA processing in *T. thermophila*.** During La-dependent processing (top, left) in previously studied eukaryotes such as yeast, the La protein is the first protein to associate with pre-tRNAs on the 3′-stretch of uridylates generated by RNA polymerase III transcription termination. Binding of La provides protection from degradation by 3′-exonucleases, assists with tRNA folding through RNA chaperone activity and stabilizes the nascent pre-tRNA. The next step in tRNA processing is endonucleolytic removal of the 5′-leader by RNase P, followed by an endonucleolytic cut by RNase Z resulting in removal of the 3′-trailer sequence and the La protein bound to the uridylate stretch. In contrast, during La-independent processing of pre-tRNAs (top, right), the 3′-trailer is rapidly removed first by 3′-exonucleases such as Rex1p, followed by RNase P processing resulting in an end-matured tRNA. Our data from *T. thermophila* point towards a pre-tRNA processing model in which Mlp1-dependent processing (bottom, left) 3′-trailers are processed efficiently. Mlp1-independent processing results in the accumulation of pre-tRNAs containing unprocessed 3′-trailer sequences, indicating that Mlp1 is required for efficient 3′-end processing unlike other eukaryotes. Image created with BioRender.com. The Publication License is provided as a Source Data file.

uridylate tail and thereby providing protection against 3′-exonucleases. Using northern blotting and RIP-3′-RACE, we demonstrated that while *S. pombe* La stabilizes 5′-leader, 3′-trailer, and intron-containing pre-tRNA intermediates, Mlp1 stabilized a 5′-leader and intron-containing, but 3′-trailer lacking, pre-tRNA species, suggesting that Mlp1 may accelerate 3′-processing of nascent pre-tRNAs, relative to other La proteins. This could imply that binding of Mlp1 to the 3′-ends of the nascent pre-tRNAs is not as strong as compared to Sla1p, leaving the RNA exposed to 3′-exonucleases. Alternatively, Mlp1 could increase access of the 3′-trailer to the endonuclease RNase Z.

To better understand how the altered architecture of Mlp1 might influence pre-tRNA processing, we generated a partial *MLP1* knockout strain in *T. thermophila* and performed RNA-Seq and northern blots of endogenous tRNA species. We found that, consistent with our Mlp1 data from *S. pombe*, Mlp1 appears to promote the removal of 3′-trailers, as depletion of Mlp1 in vivo leads to a greater relative abundance of UUU-3′OH trailer extensions. In yeast, depletion of La leads to exo-nucleolytic nibbling and a lower abundance of UUU-3′OH ends, while in the presence of La the 3′-end is stabilized until endonucleolytic cleavage by RNase Z directly 3′ to the discriminator base ($N_{73}$) (Fig. 7).

Previous work has demonstrated that hLa impedes access by RNase Z, delaying the 3′-endonucleolytic cleavage and resulting in 5′-leader processing proceeding 3′-trailer processing[43]. We hypothesize that the presence of an RRM1 in hLa could be contributing to this function, seeing that in Mlp1, a La protein lacking the RRM1, the 3′-processing appears not to be blocked. Thus, the apparent destabilization of 3′-trailers by Mlp1 in *T. thermophila* suggests that the function of La in this system, differing from other examined La proteins, may be to increase access of the pre-tRNA 3′-trailer to the tRNA processing machinery (Fig. 7). The difference in the processing of pre-tRNAs is likely caused by Mlp1 since the 3′-exonuclease Rex1p and 3′-endonuclease RNase Z are conserved in *T. thermophila* (Supplementary Fig. S9a, b). An alternate but not mutually exclusive hypothesis may be that reduction of Mlp1 levels may alter access of pre-tRNA 3′-ends to the LSm2–8 complex, and that this may be linked to 3′-trailer accumulation relative to wild-type cells, as previous work in yeast has also linked the LSm2–8 complex to the processing of pre-tRNA 3′-ends[44,45]. To confirm that the LSm2–8 complex functions similar as in other eukaryotic systems, we compared primary amino acid sequence alignments and found that the LSm2–8 complex in *T. thermophila* appears to be conserved, indicating similar functionality (Supplementary Fig. S9c, d). We also observed that Mlp1 depletion did not lead to changes in mature tRNA expression levels, suggesting that an Mlp1-independent tRNA maturation pathway also likely exists in *T. thermophila*, similar to budding and fission yeast.

Along with the alternate Mlp1-associated tRNA processing in *T. thermophila*, we observed several differences in features of pre-tRNA 5′-leaders and 3′-trailers compared to other eukaryotic species. Our

genome-wide 3′-trailer length analysis in eukaryotes revealed that *T. thermophila* pre-tRNAs have very short 3′-trailer sequences. Since 3′-trailers in *T. thermophila* are dominated by the uridylate stretch, we studied the prevalence of the 3′-most terminal residue of the 5′-leader ($N_{-1}$) and found that *T. thermophila* pre-tRNAs appear to have evolved to avoid perfect matches with the discriminator base ($N_{73}$) (preceding the uridylate tail), ensuring a lack of complete base pairing to enhance RNase P cleavage in the context of a minimized 3′-trailer sequence. Further upstream in the 5′-leader ($N_{-2}$) is often an adenosine, but then typically uridylates, which would not be expected to pair with the 3′UUU-OH motif. The 5′-leaders and 3′-trailers in *T. thermophila* may therefore have evolved to have minimal base pairing between pre-tRNA 5′-leaders and 3′-trailers, compared to other eukaryotes in which a bulge proximal to the mature tRNA ends is often followed by a paired region closer to the 5′-leader and 3′-trailer extremities. For example, human pre-tRNAs have an approximately equal distribution of $N_{-1}$ nucleotide for discriminator bases A, T and G. The most prevalent $N_{-1}$ nucleotide for discriminator bases A and T, are T and A, respectively, indicating that human pre-tRNAs, which generally contain a longer 3′-trailer sequence, have more flexibility in $N_{-1}$ sequence to introduce a mismatch bulge at the $N_{-1}$ site for optimal 5′-leader processing by RNase P. It will be interesting to further investigate whether this alternate arrangement is linked to the altered functional roles described here for Mlp1 in this system.

## Methods

### Conservation analysis

Accession numbers used to obtain primary amino acid sequences from the National Center for Biotechnology Information (NCBI) for La, Rex1p, RNase Z/ELAC 2, and LSm2−8 complex are shown in Supplementary Table S3 and primary amino acid alignments were obtained using Clustal Omega (EMBL-EBI)[46], followed by analysis using a custom python script to annotate identical and conserved amino acids using human protein sequences as a reference. Amino acids were grouped as conserved based on side chain diversity: (1) Asp (D), Glu (E), Asn (N), Gln (Q); (2) Lys (K), Arg (R), His (H); (3) Phe (F), Trp (W), Tyr (Y); (4) Val (V), Ile (I), Leu (L), Met (M); and (5) Ser (S), Thr (T).

Secondary structure predictions were obtained using Phyre2[20] and color-coded based on predicted β-sheet or α-helices. High-resolution structures of the LaM of *H. sapiens, T. brucei*, and *D. discoideum* La proteins were obtained from PDB: 2VOD, 1S29, and 2M5W, respectively. Structure of the La module (LaM + RRM1) in complex with uridylate RNA of *H. sapiens* La was obtained from PDB: 2VOD. Structure prediction of the LaM of *T. thermophila* was obtained using the lomets2 tool[47]. Structure of the LSm2−8 complex in complex with uridylate RNA of *S. pombe* and *S. cerevisiae* were obtained from PDB: 6PNN and 4M7D, respectively. Modeling was done using PyMOL.

### DNA constructs

The *MLP1* sequence encoding *T. thermophila* La protein (NCBI Reference Sequence: XP_001019287.2) was obtained from the Genome Database http://www.ciliate.org/ (TTHERM_00384860)[48], and codon optimized for expression in *Escherichia coli*. gBlocks of the optimized codon sequence were ordered from integrated DNA technologies (IDT) (sequence can be found in Supplementary Data 3).

Mlp1 and Mlp1 mutants used for in vitro electromobility shift assays (EMSAs) were cloned in the *NheI* and *BamHI* restriction enzyme sites of the pET28a vector using the plasmid-encoded N-terminal 6X-His-tag for protein purification. pET28a hLa was previously cloned in the *NcoI* and *BamHI* sites, including a reverse primer-encoded 6X-His tag. Mlp1 and Mlp1 mutants used for *S. pombe* transformations were cloned in the *SalI* and *BamHI* sites in pRep4 plasmid (ura+) incorporating a 6X-His tag N-terminally in the forward primer during PCR amplification. All primers are listed in Supplementary Data 3.

T7 DNA templates for in vitro transcription were generated by PCR amplification using pre- or mature tRNA-specific primers (listed in Supplementary Data 3) to obtain DNA templates containing an upstream T7 promotor. The DNA template was gel purified using 7 M urea denaturing 10% polyacrylamide gel and DNA was eluted from the gel by overnight rotation in 150 mM sodium acetate in 50% phenol:-chloroform:isoamylalcohol (25:24:1) at 4 °C. The aqueous layer obtained by centrifugation at $20,000 \times g$ for 10 min at 4 °C was ethanol precipitated overnight at −80 °C.

### *T. thermophila* cultivation and knockout strain generation

Liquid cultures were grown to mid-log phase ($0.1–1 \times 10^6$ cells/mL) at 30 °C shaking at 90 RPM in SPP (1% proteose peptone, 0.1% yeast extract, 0.2% glucose, 0.003% Fe-EDTA)[49]. Cell pellets were harvested by centrifugation for 3 min at $1000 \times g$. Following two washes in 10 mM Tris-HCl, pH 7.4, pellets were stored at −80 °C.

PCR amplification from *T. thermophila* genomic DNA to obtain flanking 5′- and 3′-regions was completed using primers containing *KpnI/XhoI* and NotI/SacI, respectively. The flanking regions of the *MLP1* gene (http://www.ciliate.org/-TTHERM_00384860 [http://ciliate.org/index.php/feature/details/TTHERM_00384860]) were cloned into pNeo4 plasmid[50] flanking the paromomycin (Neo4) drug resistance cassette using restriction enzyme sites *KpnI XhoI* and *SacI NotI*, respectively. The Neo4 cassette is located downstream of the $CdCl_2$ inducible metallothionein (MTT1) promoter. The resulting pNeo4 *MLP1* knockout plasmid DNA construct was linearized using the *ScaI* restriction enzyme prior to transformation. Biolistic transformation of *T.thermophila* was performed as described previously[51]. Integration of the DNA construct is based on homologous recombination and transformants were grown under increasing concentration of paromomycin starting at 60 µg/mL to a final concentration of 1000 µg/mL (phenotypic assortment)[30]. Correct integration was determined using southern blot and PCR. Partial MLP1 knockout strain available on request.

### Protein isolation from *T. thermophila*

*T. thermophila* cell pellets were resuspended in 10% trichloroacetic acid (TCA) in 1× PBS, followed by incubation at −20 °C overnight to enhance protein precipitation. The white fluffy protein pellet was collected by centrifugation for 15 min at $10,000 \times g$ at 4 °C, then washed twice with 100% ice-cold acetone. The pellet was air-dried and resuspended in 50 µL/mL culture 2.5× loading dye [5× loading dye: 5% β-mercaptoethanol (v/v), 0.02% bromophenol blue (w/v), 30% glycerol (v/v), 10% sodium dodecyl sulfate (SDS) (w/v), 250 mM Tris-HCl, pH 6.8].

### RNP-immunoprecipitation from *T. thermophila*

*T. thermophila* cell pellets from 100 mL cultures were collected at log phase ($0.1–1 \times 10^6$ cells/mL) and washed twice with 1× PBS. The cells were cross-linked with 1% formaldehyde for 10 min followed by quenching with 0.25 M glycine for 5 min at room temperature. Cells were washed twice in 1× PBS before lysis in 2 mL buffer A [30 mM Tris-HCl pH 7.4, 150 mM NaCl, 20 mM KCl, 2 mM $MgCl_2$, 0.1% Triton-X100, 1 mM phenylmethylsulfonyl fluoride (PMSF), 1× Halt Protease Inhibitor Cocktail (PIC) (ThemoFisher Scientific), 100U SUPERase In RNase Inhibitor (ThermoFisher Scientific)] followed by sonication (25% amplification with 15 s intervals for 4 min). The lysate was clarified by centrifugation at $20,000 \times g$ for 45 min at 4 °C and treated with 10 U TurboDNase. The lysates were pre-cleared using rabbit isotype immunoglobin G (IgG) control-bound Protein G Dynabeads rotating for 1 h at 4 °C. Immunoprecipitations were performed using an affinity-purified rabbit anti-Mlp1 antibody (ThermoFisher Scientific−Custom Antibodies) and a rabbit IgG control coupled to Protein G Dynabeads rotating 1 h at 4 °C. The Protein G Dynabeads were washed five times

using buffer A (without PIC, PMSF, and RNase inhibitor). Input and eluted RNA was isolated following reverse cross-linking for 45 min at 70 °C in buffer B [50 mM Tris-HCl pH 7.4, 5 mM EDTA pH 8.0, 10 mM DTT, 1% SDS], followed by Trizol extraction.

### tRNA library preparation and TGIRT sequencing and analysis

Total and Mlp1-associated RNAs were size-selected (20–300 nucleotides) on a 7 M urea-denaturing 10% polyacrylamide gel. RNA was eluted from the gel by overnight rotation in 150 mM sodium acetate, pH 5.1 in 50% phenol:chloroform:isoamylalcohol (25:24:1) at 4 °C. The aqueous layer obtained by centrifugation at $20,000 \times g$ for 10 min at 4 °C was ethanol precipitated overnight at −80 °C. The pellet was washed once with 70% ethanol, air-dried, and resuspended in RNase-free $H_2O$. To promote sequencing of highly structured tRNA species carrying modifications that are inhibitory to cDNA synthesis, we used TGIRT-III reverse transcriptase (InGex) template-switching to prepare cDNA libraries as described previously[52,53]. Libraries were sequenced on a NextSeq500 platform (Illumina) (2 × 75 bp).

### tRNA-Seq processing pipeline

Details about tools and parameters of bioinformatic analyses are provided in a reproducible Snakemake workflow that can be found at https://github.com/etiennefc/t_thermophila_RNA_Seq.git and are also described below. The raw data (.fastq files) are available for download on the Gene Expression Omnibus (GEO) under the accession number GSE199642. Briefly, paired-end reads were trimmed using Trimmomatic v0.36[54], and FastQC v0.11.5 was used to evaluate read quality before and after trimming, as described previously[52]. Resulting trimmed reads were then aligned stringently (i.e., allowing no mismatch) to the *T. thermophila* genome assembly version T_Thermophila_MAC_2021[55] (accessible on the Genome Database at http://www.ciliate.org/system/downloads/1-upd-Genome-assembly.fasta) using STAR v2.6.1a[56] (with the parameters described previously[52] and also the following parameter and value:−outFilterMismatchNmax 0. The index needed to align reads to the genome was produced using STAR v2.6.1a as described previously[52] using the genome assembly described previously and a custom *T. thermophila* annotation (.gtf) file available at https://zenodo.org/record/6391187#.YkH9v-fMK3A. The annotation file was built by converting.gff files (one for protein-coding genes, one for tRNA genes and one for 5 S ribosomal RNA genes; these.gff files are also available at https://zenodo.org/record/6391187#.YkH9v-fMK3A) into.gtf files and by concatenating these files into one final (.gtf) annotation file using custom bash scripts. This annotation was corrected for embedded genes using CoCo v0.2.5.p1[57] with the correct_annotation mode and default parameters. Counts were attributed to genes and normalized as transcripts per million (TPM) as previously described[52] using CoCo v0.2.5.p1 with the correct_count mode. Bedgraph files were generated using CoCo v0.2.5.p1 (with the correct_bedgraph mode with default parameters). Differential expression analysis was performed using DESeq2[58] with default parameters and the count output of CoCo correct_count to compare the wild type, Mlp1 IP and partial Mlp1 knockout samples. Normalized counts measured in TPM and differential expression data for tRNAs are shown in Supplementary Data 4. The reproducibility of tRNA counts in biological replicates was determined by generating correlation plots of the TPM after filtering reads with low counts in R.

### tRNA read fishing and binning into pre-tRNAs and mature tRNAs

Raw counts for pre-tRNA (3′-UUU) and mature tRNAs (3′-CCA) were generated using custom python scripts. A list of unique sequences was generated for each tRNA isoacceptor (e.g., Gln$^{UUG}$: AATCCTCT GACCTGGGTTCGAATCCCAGTACGACCT) (Supplementary Data 5) and used to obtain ("fish") all reads from the unmapped raw sequence

files (.fastq file format). Each sequence was grouped in corresponding bins based on the 3′-end of the reads: -CCA (mature tRNA), 1U, 2U, 3U, 4U, 5U, 6U, 7U, 8U, 9U, or 10U (premature tRNA). Raw counts for each tRNA isotype were normalized as counts per million (CPM) by dividing raw counts by the total number of fished read per bin for each replicate divided by $10^6$ (Supplementary Data 6 and Supplementary Data 7). Fold enrichment for Mlp1-bound tRNAs was calculated after summation of pre-tRNA counts, followed by taking the ratio of CPM data for Mlp1-immunoprecipitated tRNAs and wild type input tRNAs and displayed after log2 transformation in heatmaps (Supplementary Data 6). Fold enrichment for hLa-bound tRNAs from ref. 22 was obtained from the GEO database under accession number GSE95683 where counts for pre-tRNAs and mature tRNAs were normalized as CPM and transformed identically as Mlp1-bound tRNAs. The cumulative abundance in CPM for each pre-tRNA isoacceptor was calculated for wild type and partial Mlp1 knockout tRNAs and displayed after log10 transformation in heatmaps (Supplementary Data 7).

### Electromobility shift assays (EMSA)

U10 and CUGCUGUUU (20 pmol) were chemically synthesized (Integrated DNA Technologies (IDT)) and 20 pmol RNA was radioactively labeled using [γ-$^{32}$P]-ATP (PerkinElmer, 10 mCi/ml) and 10 units T4 Polynucleotide Kinase (PNK) enzyme (New England Biolabs, cat#M0201S) for 2 h at 37 °C in 1× T4 PNK buffer (New England Biolabs, cat#B0201S). Radioactively labeled tRNAs were produced by T7 in vitro transcription in the presence of [α-$^{32}$P]-UTP (PerkinElmer, 10 mCi/ml) using PCR products containing an upstream T7 promotor (see DNA constructs). Dephosphorylation of the 5′-triphosphate pre-tRNA was done using 5 units QuickCIP (New England Biolabs, cat#M0525S) for 30 min at 37 °C in 1× rCutSmart Buffer (New England Biolabs, cat# B6004S), followed by Trizol extraction. All radioactively labeled RNAs were purified on a denaturing polyacrylamide gel and eluted in 0.5 M NaCl overnight at room temperature.

His-tagged proteins hLa, Mlp1 and Mlp1 mutants were purified from *E. coli* BL21 cells using $Co^{2+}$ beads, followed by heparin column purification. The proteins were buffer exchanged in 1× PBS and quantified using Bovine Serum Albumin (BSA) quantifications on SDS-polyacrylamide gel.

EMSAs were performed as described[24]. Briefly, 3000 CPM of radioactive RNA substrates (~ 0.1 nM) were incubated in 1X EMSA buffer [10% glycerol, 20 mM Tris-HCl, pH 7.4, 100 mM KCl, 1 mM EDTA, 5 mM β-mercaptoethanol and bromophenol blue] at 95 °C for 5 min, followed by slow cooling to room temperature. For competition EMSAs, unlabeled RNA was added to the radioactively labeled RNA prior to incubation at 95 °C. The concentration of unlabeled RNA depends on the protein concentration used to bind >85% of the radioactively labeled RNA. Protein was added and incubated for 30 min at 30 °C. The protein–RNA complexes were immediately snap cooled on ice for 5 min and separated on a 8% native polyacrylamide gel at 4 °C at 100 V. The gels were dried for 45 min at 80 °C on a Gel Dryer (Bio-Rad) and exposed to a storage phosphor screen overnight. The screens were developed on a Typhoon. Quantification of bound and free RNA was done using ImageJ and binding curves were fit using a nonlinear-specific binding curve fitting program and Kd values were calculated (GraphPad Prism).

### tRNA-mediated suppression assay in *S. pombe*

tRNA-mediated suppression assays were performed as described previously[27]. In brief, the *sla1- S. pombe* ySH9 strain encoding a defective UGA-decoding suppressor tRNA (tRNA-Ser$^{UCA}$) and the *ade6-704* allele was transformed using a pRep4 plasmid encoding Sla1p, Mlp1 and multiple Mlp1 mutants. Following the transformation of *S. pombe* suppressor strains using pRep4 plasmid (*ura4+*), strains were grown on selective media (Edinburgh Minimal Medium (EMM) −ura −leu) and grown in liquid EMM −ura − leu to mid-log phase (OD

0.6–0.9). Spotting was done by transferring 4 µL of liquid culture on low adenine-containing plates (EMM –ura –leu ade10), followed by a 4-day incubation at 32 °C. Yeast pellets for protein purification or RNA extraction were obtained by centrifugation of mid-log phase cells at 1800 × g for 10 min followed by two washes with ddH$_2$O.

Protein extraction was done by resuspension of cell pellets in NET-2 buffer [50 mM Tris-HCl, pH 7.4, 150 mM NaCl, 0.05% NP40, 1 mM PMSF, and 1× PIC], followed by lysis through bead-beating for 2 min total (20 s ON – 20 s OFF intervals) at 4 °C. Cell lysates for protein analysis were obtained following centrifugation for 15 min at 20,000 × g.

RNA extraction was completed by resuspending the cell pellets in complete RNA extraction buffer A [50 mM NaOAc, pH 5.1, 10 mM EDTA, 1% SDS], followed by adding 37 °C buffer A [50 mM NaOAc, pH 5.1]-saturated acid phenol and incubation at 65 °C for 4 min with frequent vortexing. The aqueous top layer was extracted following centrifugation for 3 min at 20,000 × g and extracted again using phenol:chloroform:isoamylalcohol (25:24:1). RNA was precipitated from the aqueous layer by ethanol precipitation and incubation at −80 °C for at least 1 h.

### RNA-immunoprecipitation and 3′-RACE in *S. pombe*

Sla1p- and Mlp1-transformed *S. pombe sla1-* strains (ySH9) were grown to mid-log phase (OD 0.6–0.9) in EMM –ura –leu. The culture was cross-linked at 200 RPM in 0.5% formaldehyde at room temperature for 20 min, followed by adding 200 mM glycine for 10 minutes. The yeast pellet was collected by centrifugation for 10 min at 4000 RPM (Beckman Coulter JLA9.1000 rotor)/3399 × g, washed with ddH$_2$O and collected by centrifugation for 10 min at 1800 × g, followed by one wash in resuspension buffer [1.2% polyvinylpyrrolidone (PVP), 20 mM HEPES, pH 7.4, 1 mM PMSF, 1× PIC, 1 mM DTT]. The yeast pellet was flash frozen in liquid nitrogen as a continuous bead, followed by cryogrinding in liquid nitrogen using a mortar and pestle. Yeast powder was lysed in RNP buffer [20 mM HEPES, pH 7.4, 110 mM KOAc, 100 mM NaCl, 0.5% Triton-X100, 0.1% Tween-20, 1 mM PMSF, 1× PIC and 0.05 U/µL RNase inhibitor]. The lysate was clarified by centrifugation at 20,000 × g for 10 min at 4 °C. Next, 0.005 U/µL TurboDNase (ThermoFisher Scientific AM2238) was added, and the cell lysate was pre-cleared using Protein G Dynabeads coated with rabbit isotype immunoglobin G (IgG) control rotating 1 h at 4 °C. Immunoprecipitations were performed using an affinity-purified rabbit anti-Mlp1 and anti-Sla1p antibody (ThermoFisher Scientific – Custom Antibodies) coated to Protein G Dynabeads rotating 1 h at 4 °C. As a control, the same antibodies were used for immunoprecipitations from ySH9 transformed with empty pRep4 plasmid. The Protein G Dynabeads were washed 5 times using RNP buffer (without PIC, PMSF and RNase inhibitor). RNA was isolated following reverse cross-linking for 45 min at 70 °C in buffer B [50 mM Tris-HCl pH 7.4, 5 mM EDTA pH 8.0, 10 mM DTT, 1% SDS], followed by Trizol extraction.

RNA samples were polyadenylated and reverse transcribed into cDNA using qScript® microRNA cDNA Synthesis Kit (QuantaBio). Using a pre-tRNA intron-specific forward primer (Supplementary Data 3), the substrate of interest was PCR-amplified using Taq DNA polymerase in combination with the reverse PerfeCTa Universal PCR Primer (QuantaBio) which anneals with the oligo-dT adapter sequence incorporated during cDNA synthesis. The PCR products were purified through a 1% agarose gel and ligated into a pGEM-T Easy Vector Systems (Promega) plasmid followed by the transformation in *E. coli* cells. Plasmid DNA was extracted, and sequences determined by Sanger sequencing at the Hospital for Sick Children – The Centre for Applied Genomics (TCAG).

### tRNA 5′-leader and 3′-trailer computational analysis

A custom python script was used to scrape tRNA information for *H. sapiens, M. musculus, D. melanogaster, A. thaliana, S. cerevisiae* and *S. pombe* from the Genomic tRNA Database (GtRNAdb)[59] and *T. thermophila* sequences were obtained from the UCSC Genome Browser (https://genome.ucsc.edu/) (Supplementary Data 2). The number of tRNA genes encoded in the genome was determined for different eukaryotes for each isotype and isoacceptor (Supplementary Data 1). Trailer lengths were calculated as the number of nucleotides found between the discriminator base, the last annotated mature tRNA nucleotide, and a stretch of minimum four Ts in the genomic DNA. Trailer lengths longer than 20 nucleotides were excluded from the analysis. Mature tRNA sequence lengths were determined starting at nucleotide +1 and ending at discriminator base N73, excluding introns (official tRNA numbering as described previously[60]). Statistical significance ($P < 0.05$) was calculated using a one-way ANOVA and Tukey's multiple comparison test. The 5′-leader logo were generated for *H. sapiens, S. pombe* and *T. thermophila* using WebLogo[61] and divided based on discriminator base identity.

### Immunofluorescent staining in *T. thermophila*

*T. thermophila* wild type and partial *MLP1* knockout strains were grown to mid-log phase and prepared for indirect immunofluorescent staining as described[62]. Centrifugation steps, including washes, were performed for 3 min at 1000 × g. Briefly, fixative (two parts saturated HgCl$_2$ and one part 95% ethanol) was added to the cell suspension and incubated for 5 min at room temperature, followed by collecting and resuspending the cell pellet in 100% ice-cold methanol twice. The cell pellet was washed with PBS and incubated with primary rabbit anti-Mlp1 antibody (1:500 dilution) rotating overnight at 4 °C. The cell pellet was washed three times with PBS, followed by incubation with secondary goat anti-rabbit IgG (1:10,000, ThermoFisher Scientific A11008), rotating for 1 h at room temperature. The cell pellet was washed three times with PBS. The cell suspension was dropped on a coverslip, air-dried, and mounted with Vectashield Antifade Mounting Medium with DAPI (Vector Laboratories) onto a microscope slide. Microscopy images were obtained at 63x magnification on a LSM700 confocal laser scanning microscope (Zeiss) and processed in ZEN3.3 (blue edition).

### Northern blotting

RNA was obtained from storage solution at −80 °C [100% ethanol slurry containing 150 mM NaOAc, pH 5.1, and 30 µg GlycoBlue Coprecipitant (ThermoFisher Scientific AM9515)] by centrifugation at 20,000 × g for 10 min. The RNA pellet was washed with 70% ethanol and air-dried, followed by resuspension in RNase-free ddH$_2$O. Samples were prepared by adding an equal volume of 2X formamide loading dye [80% deionized formamide, 0.06% (w/v) bromophenol blue, 0.06% (w/v) xylene cyanol, 10 mM EDTA, pH 8.0], followed by incubation at 95 °C for 5 min and snap cooling on ice. RNA samples were separated on a 7 M urea-denaturing polyacrylamide gel at 4 °C at 100 V and transferred onto a charged nylon transfer membrane (PerkinElmer), followed by cross-linking using a UV Stratalinker and drying for 15 min at 80 °C on a Gel Dryer (Bio-Rad).

The membranes were hybridized for 2 h at the corresponding probe melting temperature (Tm) −10 °C in hybridization buffer [6× SSC (1× SSC: 150 mM NaCl, and 15 mM sodium citrate), 1% sodium dodecyl sulfate (SDS) and 4X Denhardt's solution (Bio Basic D0062)], followed by overnight incubation with T4 Polynucleotide Kinase (PNK) $^{32}$P-labeled probes (Supplementary Data 3). Following three 20-minute washes in wash buffer [2× SSC and 0.1% SDS], the membrane was exposed to a storage phosphor screen overnight and developed on a Typhoon. To strip hybridized probe, membranes were incubated three times with stripping buffer [0.1× SSC and 0.1% SDS] for 20 min each at 70 °C.

### Western blotting

Protein concentration was obtained using Bradford assays (ThermoFisher Scientific, cat#23300). Protein samples were resuspended in

1× loading dye [5× loading dye: 5% β-mercaptoethanol (v/v), 0.02% bromophenol blue (w/v), 30% glycerol (v/v), 10% sodium dodecyl sulfate (SDS) (w/v), 250 mM Tris-HCl, pH 6.8] and incubated at 95 °C for 10 min, separated by electrophoresis on a 12% SDS polyacrylamide gel and transferred onto a nitrocellulose membrane. The membrane was blocked in 0.5% (w/v) skim milk powder in Tris-Buffered Saline [20 mM Tris-HCl pH 7.4, 150 mM NaCl] + 0.1% Tween-20 (TBST) for 1 h at room temperature (or overnight at 4 °C), followed by incubation with primary antibodies in TBST for 1 h at room temperature (or overnight at 4 °C). The membrane was washed five times with TBST, followed by incubation with HRP-conjugated secondary antibodies at 1:10,000 dilutions and incubated for 1 h at room temperature. Primary antibodies used in this study: mouse anti-beta actin (Abcam ab8224), rabbit anti-histone H3 (Abcam ab1791), mouse anti-His (Abcam ab18184), and affinity-purified antibodies (ThermoFisher Scientific− Custom Antibodies) rabbit anti-Mlp1 and rabbit anti-Sla1p. Secondary HRP-coupled antibodies used in this study: goat anti-rabbit IgG (Cell Signaling Technology 7074) and horse anti-mouse IgG (Cell Signaling Technology 7076).

### Genomic DNA extraction and southern blotting in *T. thermophila*

Genomic DNA extraction from *T. thermophila* cell pellets was performed as described previously[49]. Briefly, cell pellets from 25 mL cultures were harvested and resuspended in 0.5 mL 10 mM Tris-HCl, pH 7.4, followed by the addition of 3.5 mL urea buffer [42% (w/v) urea, 350 mM NaCl, 10 mM Tris, pH 7.4, 10 mM EDTA, 1% SDS, 0.1 mg/mL Proteinase K] and incubated for 5 minutes at 50 °C. DNA was extracted twice with an equal volume of phenol:chloroform:isoamylalcohol (25:24:1), followed by chloroform:isoamylalcohol (24:1) extraction. One-third volume of 5 M NaCl was added to the aqueous phase and the DNA was precipitated with an equal volume of isopropanol. The DNA was pelleted by centrifugation at 20,000 × *g* for 10 min at 4 °C and the DNA pellet resuspended in 50 µL Tris-EDTA (TE), pH 8.0 [10 mM Tris-HCl, pH 8.0, 1 mM EDTA, pH 8.0]. The DNA suspension was treated with RNase A (10 mg/mL) overnight at 55 °C and stored at −20 °C.

Genomic DNA from wild type and partial Mlp1 knockout strains were digested with *EcoRI* restriction enzyme overnight at 37 °C and separated by electrophoresis through a 1% agarose gel. The DNA was transferred onto a nitrocellulose membrane by capillary forces, followed by cross-linking using a UV Stratalinker and drying at 80 °C on a Gel Dryer (Bio-Rad). The membrane was probed using a T4 Polynucleotide Kinase (PNK) $^{32}$P-labeled PCR product as described in "Northern blotting".

### Data reproducibility and statistics

Mlp1 RIP northern blot and associated western blot analysis were performed in biologically independent triplicates. Mlp1 partial KO strain northern blots and associated western blots were performed in biologically independent triplicates. TGIRT sequencing of size-excluded Mlp1-RIP, wild type, and Mlp1 partial knockout strains were performed in biologically independent triplicates. EMSAs and competition EMSAs were performed in at least independent experimental duplicates unless stated otherwise in figure legends. tRNA-mediated suppression assays in *S. pombe* ySH9 strains and associated northern blots and western blots were performed in biologically independent triplicates unless stated otherwise in the figure legend. RNA-immunoprecipitation and associated western blots from ySH9 and Sanger sequencing of clonal isolates derived from 3′-RACE were performed in biologically independent duplicates. Statistical analysis was performed on data derived from three or more independent replicates. Error bars represent the mean +/− standard deviation (S.D.) of at least three independent replicates. Comparison of tRNA features in different eukaryotes was done using the one-way ANOVA for 3′-trailer length ($F = 289.3$, DF = 6,

$P$ value < 0.0001) and mature tRNA length ($F = 0.9931$, DF = 6, $P$ value = 0.428).

### Reporting summary

Further information on research design is available in the Nature Portfolio Reporting Summary linked to this article.

## Data availability

TGIRT sequencing has been deposited to the Gene Expression Omnibus (GEO) under the accession number GSE199642. Previously published PDB structures used in this study: https://www.rcsb.org/structure/2VOD; https://www.rcsb.org/structure/1S29; https://www.rcsb.org/structure/2M5W; https://www.rcsb.org/structure/6PNN; https://www.rcsb.org/structure/4M7D. NCBI Mlp1 protein sequence: https://www.ncbi.nlm.nih.gov/protein/XP_001019287.2/; MLP1 reference gene: http://ciliate.org/index.php/feature/details/TTHERM_00384860. Source data are provided with this paper.

## Code availability

Code for the analysis of TGIRT sequencing and tRNA read fishing and binning into pre-tRNAs and mature tRNAs available in a Snakemake workflow that can be found at https://github.com/etiennefc/t_thermophila_RNA_Seq.git. Code for scraping tRNA sequences from the tRNA database http://gtrnadb.ucsc.edu/ is available at https://github.com/brunohenderyckx/trna-database-scraper.

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

## Acknowledgements

We thank Bruno Henderyckx for assistance with Python and Richard J. Maraia and Jean-Marc Deragon for comments on the manuscript. M.A.B. was funded by a project scheme grant from the Canadian Institutes of Health Research's (CIHR's) Institute of Genetics (application number 419907). R.E.P. was supported by CIHR [MOP13347] and NSERC Discovery Grant [539509]. M.S.S. is supported by a senior research professorship from the Fonds de Recherche du Québec-Santé and an NSERC discovery grant (RGPIN-2018-05412). S.A.E. holds a Canada Research Chair in RNA Biology and Cancer Genomics and is supported by a grant from CIHR.

## Author contributions

K.K. performed most experiments. J.G. and R.E.P. provided *Tetrahymena thermophila* wild type strains, generated the Mlp1 partial knockout strain, and associated confirmation (southern blot and PCR). S.A.E. performed cDNA TGIRT library preparation and sequencing and E.F-C., M.S.S. the associated bioinformatic analysis. K.K and M.A.B. designed the study, analyzed the data, and wrote the paper.

## Competing interests

The authors declare no competing interests.
