## [Peer Review File · Nature Communications]

Altered tRNA processing is linked to a distinct and unusual La protein in *Tetrahymena thermophila*Editorial Note: Parts of this Peer Review File have been redacted as indicated to remove third-party material where no permission to publish could be obtained.

REVIEWER COMMENTS

Reviewer #1 (Remarks to the Author):

The MS entitled "Altered tRNA processing is linked to a distinct and unusual La protein in *Tetrahymena*" by Kerkhofs et al characterize a function of unusual La protein, Mlp1, found in *T. thermophila*. The La proteins are associated with 3' end of newly synthesized small RNAs. The Introduction part is well written and stresses the major difference in pre-tRNA processing pathways. The newly characterized Mlp1 is missing a RRM1 motif adjacent to the La motif but contains a domain of unknown function-3223 (DUF3223). Interestingly it still operates as genuine La protein which was demonstrated by EMSA assays and using the *S.pombe* system. Analysis of the heterologous system, where Mlp1 was expressed in fission yeast, revealed an unclear role in the processing of the 3' trailer that could be interpreted as promotion of the 3' end shortening. Finally, the authors performed comparative analysis of tRNA 5' leaders and 3' trailers and demonstrated that *T. thermophila* has uniquely short 3' trailers. The final scheme would suggest that Mlp1 blocks RNase P cleavage whereas previously known La would favour RNase P cleavage.

The work is certainly novel, but is missing a clear message. It combines (1) biochemistry attempting to explain how novel Mlp1 works on molecular level and (2) pre-tRNA processing assays to explain its involvement in tRNA processing. In my opinion the work would benefit from stressing the message on one of the mentioned points.

Major points:

1. In the introduction the authors marked the role of La proteins in tRNA folding. La proteins can be involved in this event, however the tRNA structure itself leads to effective folding. Also Ref 1 Fairley et al do not describe a role of La in tRNA folding as it is suggested in the MS. The authors may consider that the RNA chaperoning function of La could be linked to either RNA folding or interaction/competition with surveillance machinery such as 3'->5' exonucleases.
2. Mlp1 is likely essential, but why? There is no effect on mature tRNA levels (Fig. 5G). Mlp1 in *S.pombe* does not present obvious protection of the 3' ends (Fig. 4D) and *T.thermophila* has very short or absent the 3' trailers (Fig. 6). The authors speculate that Mlp1 may promote the shortening of the 3' ends. Simultaneously Mlp1 presents RNA chaperone activity (Fig 4A). The interplay between processing/surveillance machinery is an obvious follow up for this study. One possibility to explore these questions is Co-IP of Mlp1 followed by mass-spectrometry. That could be applied to at least in the *S. pombe* system.
3. The reproducibility of sequencing data (Fig. 2B) has to be supported by statistical analysis.
4. Very interesting, but insufficiently commented, are the results from the sanger sequencing data (Fig 4D). Binding of tRNA with CCA at the 3' ends would suggest final maturation step. What Mlp1 expression in the *S. pombe* system is doing?

Minor points:

1. Shape of the binding profiles for Mlp1 (Fig 2E) may be interpreted as binary mode of binding (yes or no) when compared to hLa (Fig 2D). Could this be interpreted as interaction using one RNA-binding domain (Mlp1) vs. two RNA-binding domains (hLa)?
2. lines 176-181 sentence duplication
3. Panel 3C is too small
4. Performing analysis of the 3' end trailers the authors should be aware that transcription termination signals have different strength in different species: i.e. human terminators are typically shorter than yeast.
5. Fig S7A-B – should be presented as a boxplot including statistics

Reviewer #2 (Remarks to the Author):

In the presented manuscript, the authors functionally characterize the *Tetrahymena thermophila* protein Mlp1, that has previously been classified as a genuine La protein. Members of this protein family bind to the UUU-3'OH containing pre-tRNAs and coordinate the process of 5'-leader and 3'-trailer removal by specific RNAses (RNase P and RNase Z). In addition, the presence of La proteins is known to support pre-tRNA folding and maturation. As the ThMlp1 sequence does not display an obvious RRM, it was not clear, if Mlp1 acts as a genuine La protein and if the mechanism would be similar to its known homologues.

The authors perform comprehensive analyses, using a variety of experimental approaches. The study is well executed and the scientific claims are fully supported by the presented results. The used methodology is well documented and I could not detect any obvious signs of misinterpretations and/or overstatements. The manuscript adequately describes and discusses the available literature in the field. In detail, the authors show that Mlp1 indeed functions as La protein, binds pre-tRNAs, regulates pre-tRNA processing and supports tRNA folding. Furthermore, the authors discovered unique mechanistic features, which can be associated with the presence of unusually short 3'-trailers in *Tetrahymena* – a characteristic that the authors also describe for the first time. Therefore, the study indeed presents an appealing and novel mechanisms of pre-tRNA processing that is coordinated by Mlp1. The work might provide a reference study that allows to discriminate between canonical and non-canonical routes of pre-tRNA processing in the future. Hence, I in principle support the publication in *Nature Communication*, but I would suggest to use state-of-the-art structure prediction approaches to present more complete structural models of full length Mlp1.

Major issue

I would suggest to use alphafold2 (or similar algorithms) to predict the structure of Mlp1 de novo. The presented homology model is obviously already very useful to identify crucial residues in Mlp1, but considering the low sequence conservation it is not surprising that it falls short in providing information about other regions of the protein. Considering the presented biochemical results, I could imagine that a cryptic RRM domain could actually be present in ThMlp1. In addition, it would be useful for the reader to have at least a tentative model of the DUF3223 domain, which also could present RNA recognition surfaces as it seems to be important for the described RNA chaperoning function as well.

Minor issues

- Please introduce Mlp1 in the first paragraph of the results section – otherwise this paragraph reads very confusing
- Please remove the redundant “nM” labels for the truncated constructs in Table 1

Reviewer #3 (Remarks to the Author):

In this manuscript, Kerkhofs et al study the mechanism of Mlp1, a La-like protein in *T. thermophila* that seems to confer to the species an alternative La protein dependency in tRNA maturation as compared with other eukaryotes. They conclude that: 1) Mlp1 is likely a La protein, 2) Mlp1 has lower affinity for pre-tRNAs and their 3' trailer sequences than human La protein, 3) Mlp1 has a tRNA chaperone function, and 4) Mlp1, unlike other RNA recognition motif-1 (RRM1)-containing La proteins, destabilizes tRNA 3' trailer sequences. This work provides a useful characterization of an otherwise poorly understood mechanism in *T. thermophila* and provides evidence for proper interpretation of tRNA maturation in species lacking genuine characterized La proteins. Nonetheless, I have some concerns regarding data and conclusions presented in the manuscript, as outlined below:

1. In Figure 2A/B, the authors demonstrate that both Mlp1 and hLa have a binding preference for pre-tRNAs over mature tRNAs. While the data is compelling, the authors should comment on why there is

increased variability in the enrichment of pre-tRNAs when looking across isoacceptor classes. It seems that hLa has high preference for all pre-tRNAs except for Cys tRNAs, for which there is no preference for pre- over mature species. There are several isoacceptor classes for which Mlp1 does not exhibit preference for the pre-tRNA. Could it be inherent to the trailers of these classes? It seems that the preference correlates with the amino acids in an alphanumeric manner, and is this some sort of technical artifact in the data?

2. The experimental design for results presented in Figure 3A/B could be clarified. Are human or tetrahymena mature and pre-tRNAs used for the competition assay? Are these a mixture of all tRNA species or enriched for particular isodecoders?

3. It is unclear why the U>C change is made in the 4U construct used to generate Figure 3D/E. Why not change U to A or G, and would this be expected to impact the results shown here? The authors should demonstrate that there is no nucleotide preference for the phenomenon described when comparing hLa and Mlp1 data.

4. The illustrations adjacent to Figure 4B/C are, in certain cases, difficult to map to the bands of the northern blots. Perhaps some sort of boxing around the bands could be done to improve this. That said, the identities of all of the bands on these blots is not always clear with the provided illustrations, especially in Figure 4C. There are bands for which there are no corresponding illustrations, leading to difficulty in independent interpretation of the blots. The authors should fix this to provide comprehensive annotation where possible. Furthermore, it is unclear how the authors mapped each of the illustrations to their corresponding bands. The authors should provide additional evidence by means of 3' and 5' trailer probes and/or 5' and 3' processing defective strains to confirm the identities of their northern blot bands.

5. It seems to be the case, with the northern blot in Figure 4C, that Mlp1 stabilizes tRNAs with a fully intact 3' trailer sequence, as indicated by the adjacent illustrations. The authors should discuss this inconsistency in light of the conclusions made in this paper that limiting Mlp1 expression stabilizes 3' trailers.

6. The authors should discuss the presence of mature tRNA sequence that seems to have been processed in Figure 4D. What is the nature of this loss of sequence in the Mlp1-bound Lys and Tyr tRNAs?

Minor edits:

1. 'suppressor tRNA stabilization' is indicated on Figure 4C. I think that this should not be indicated as a suppressor tRNA.
2. It is somewhat difficult to evaluate the effectiveness of suppression between some of the colonies in Figure 4A due to differing background color tone. The difference between + and ++ is not clear to me.

We thank the reviewers for their comments and for consequently improving our manuscript submission. We address their comments below in a point-by-point fashion.

REVIEWER COMMENTS

Reviewer #1 (Remarks to the Author):

The MS entitled “Altered tRNA processing is linked to a distinct and unusual La protein in Tetrahymena” by Kerkhofs et al characterize a function of unusual La protein, Mlp1, found in T. thermophila. The La proteins are associated with 3’ end of newly synthesized small RNAs. The Introduction part is well written and stresses the major difference in pre-tRNA processing pathways.

The newly characterized Mlp1 is missing a RRM1 motif adjacent to the La motif but contains a domain of unknown function-3223 (DUF3223). Interestingly it still operates as genuine La protein which was demonstrated by EMSA assays and using the S.pombe system. Analysis of the heterologous system, where Mlp1 was expressed in fission yeast, revealed an unclear role in the processing of the 3’ trailer that could be interpreted as promotion of the 3’ end shortening. Finally, the authors performed comparative analysis of tRNA 5’ leaders and 3’ trailers and demonstrated that T. thermophila has uniquely short 3’ trailers. The final scheme would suggest that Mlp1 blocks RNase P cleavage whereas previously known La would favour RNase P cleavage.

The work is certainly novel, but is missing a clear message. It combines (1) biochemistry attempting to explain how novel Mlp1 works on molecular level and (2) pre-tRNA processing assays to explain its involvement in tRNA processing. In my opinion the work would benefit from stressing the message on one of the mentioned points.

We thank the reviewer and are pleased that they found our introduction well written and the study novel. It has been indeed challenging to combine the biochemical and pre-tRNA processing narratives in our work, but given the extensive study of La structure/function (encapsulated in the La module-UUU’3’OH co-crystal structure, Teplova et al., 2006 *Mol Cell*) and the well-documented role of La in pre-tRNA function (primarily Yoo and Wolin, 1997, *Cell*), we feel the onus is on us to ensure that both of these fundamental narratives are addressed in a work that examines a new class of La protein.

Upon reflection, we feel it is the variant La protein which is the original impetus for the work, and that the consequent effects on pre-tRNA processing subsequently emerge from this original theme. We have thus added new sentences to the manuscript that better articulate how our computational and biochemical work (Figures 1-3) follow through to the work in which we investigate pre-tRNA processing (Figures 4 & 5). Specifically:

Introduction, page 4, new text in italics: “Using ribonucleoprotein immunoprecipitation (RIP)-Seq of Mlp1, we show association with UUU-3’OH containing pre-tRNAs *in vivo*, and

preferential binding of pre-tRNA substrates over mature tRNA substrates *in vitro*. *In order to assess whether this altered architecture might have associated consequences in pre-tRNA processing, we tested Mlp1 function in a well-established model system and demonstrate that heterologous Mlp1 expression in Schizosaccharomyces pombe promotes pre-tRNA processing and tRNA mediated suppression, but without typical La-associated 3'-end protection...*”.

Results, page 11: *“Our computational and biochemical work suggested that Mlp1 differs from other examined La proteins in its binding and discrimination of RNA targets. We and others have previously used a well-established, heterologous tRNA-mediated suppression system in S. pombe to investigate the function of both yeast and human La in the promotion of La-dependent pre-tRNA processing. This system can report on La-dependent 3'-end protection of nascent pre-tRNAs from exonucleases, as well as RNA chaperone activity to enhance correct folding of nascent pre-tRNAs, through the ability to rescue a misfolded suppressor tRNA in vivo. To test whether the altered Mlp1 architecture correlated with differences in pre-tRNA processing, we transformed Sla1p, full-length Mlp1 and multiple Mlp1 mutants...”*

Furthermore, in alignment with the point raised by Reviewer #2, we have also made reference to Mlp1 in the first paragraph of the Results section, so as to establish the primary importance of the work on the investigation of this protein.

Altogether, we feel the revised text establishes the primacy of the variant La protein first, but now also better segues into the consequent effects on pre-tRNA processing in a way that each narrative mutually reinforces the other.

Major points:

1. In the introduction the authors marked the role of La proteins in tRNA folding. La proteins can be involved in this event, however the tRNA structure itself leads to effective folding. Also Ref 1 Fairley et al do not describe a role of La in tRNA folding as it is suggested in the MS. The authors may consider that the RNA chaperoning function of La could be linked to either RNA folding or interaction/competition with surveillance machinery such as 3'->5' exonucleases.

We have moved the Fairley reference in the manuscript so as to more accurately reflect the intended nature of this citation (La is the first factor to bind pre-tRNAs), and have also added more appropriate references for the purpose of detailing La function in pre-tRNA folding (Chakshusmathi et al. 2003; Copela et al., 2006 and Bayfield & Maraia, 2009) as well as La interaction/competition with 3' exonucleases and nuclear surveillance (Huang et al., 2006 & Copela et al., 2008).

2. Mlp1 is likely essential, but why? There is no effect on mature tRNA levels (Fig. 5G). Mlp1 in S.pombe does not present obvious protection of the 3' ends (Fig. 4D) and T.thermophila has very short or absent the 3' trailers (Fig. 6).

La is essential in a number of systems including fruit fly, mice and human cells, but the reason is not clear. Inducible knockout (Cre/Lox) of La mouse B cells or the mouse forebrain leads to rapid depletion of B cells or brain cell mass, respectively, with consequence changes in pre-tRNAs but minimal changes to mature tRNA levels (Gaidamakov et al., *MCB*, 2014), similar to what is described in the current work. We have revised the manuscript to emphasize this point on page 15.

The authors speculate that Mlp1 may promote the shortening of the 3' ends. Simultaneously Mlp1 presents RNA chaperone activity (Fig 4A). The interplay between processing/surveillance machinery is an obvious follow up for this study. One possibility to explore these questions is Co-IP of Mlp1 followed by mass-spectrometry. That could be applied to at least in the S. pombe system.

We have indeed initiated this work, in which we have immunoprecipitated Mlp1 from *T. thermophila* cells followed by identification of candidate interacting proteins by LC-MS/MS. These are currently undergoing follow up/validation and, if successful, will form the basis for future work.

3. The reproducibility of sequencing data (Fig. 2B) has to be supported by statistical analysis.

We have determined the reproducibility of the replicates by plotting the reads of the replicates against one another and have provided this information in a revised **Supplementary Figure 3**.

4. Very interesting, but insufficiently commented, are the results from the sanger sequencing data (Fig 4D). Binding of tRNA with CCA at the 3' ends would suggest final maturation step. What Mlp1 expression in the S. pombe system is doing?

The RIP-Sanger sequencing results in Fig 4D rely on a cDNA amplification step that includes a forward primer in the pre-tRNA intron sequence, thus ensuring that we are detecting pre-tRNA species in this assay. Since CCA addition occurs in the nucleus before nuclear export and pre-tRNA intron splicing (Wolfe et al., 1996; reviewed in Hopper and Phizicky, 2010), these results suggest that Mlp1 is capable of binding nuclear pre-tRNA species that have been 3' end processed (degraded from the 3' end or 3'CCA added), consistent with our other data indicating diminished 3' end discrimination by Mlp1 relative to hLa (Figure 2) and Sla1p (Figure 4A-C). We have added a new sentence to the manuscript on page 13 to highlight this point and thank the reviewer for helping make the manuscript consequently clearer.

Minor points:

1. Shape of the binding profiles for Mlp1 (Fig 2E) may be interpreted as binary mode of binding (yes or no) when compared to hLa (Fig 2D). Could this be interpreted as interaction using one RNA-binding domain (Mlp1) vs. two RNA-binding domains (hLa)?

It is possible that the different shapes of the curves may reflect something like this but we feel it would be speculative to raise this point at this time. Along these lines, we note that deletion

of the La motif abolishes UUU-3'OH binding, and the La motif by itself is incapable of UUU-3'OH binding (Supplementary Figure S4b), similar to what would be expected for human La, despite the lack of the RRM. We are currently performing further structure-function analysis on Mlp1 (see point for reviewer 2, below) and so we are hopeful that a more comprehensive future analysis may shed insight here.

2. lines 176-181 sentence duplication

This has been resolved; we thank the reviewer for catching this.

3. Panel 3C is too small

Figure 3C has been enlarged.

4. Performing analysis of the 3' end trailers the authors should be aware that transcription termination signals have different strength in different species: i.e. human terminators are typically shorter than yeast.

For the trailer length analysis in the manuscript we presumed that terminators in *T. thermophila* contained at least 4 consecutive Ts in the non-template strand. While no work that we are aware of has tested *T. thermophila* RNA polymerase III for its T-length terminator requirements, we have noted in our lab that Mlp1 has equal affinity for a U10 RNA and the CUGCUGUUUU 10-mer, which ends in 4Us. Since La uridylate binding requirements have been previously shown to correlate with RNA polymerase III terminator requirements (Hamada et al., 2000 JBC and Huang et al., 2005 MCB), this represents indirect evidence that the 4T length is a good approximation of terminator requirements in *T. thermophila*.

Based on this comment, however, to test this further we reran the trailer length analysis but allowing for a 5T, 6T or 7T terminator requirement. We found that *T. thermophila* trailers were consistently the shortest of any examined species for all tested T lengths:

Average trailer lengths: 4, 5, 6 or 7T terminators

Species	Termination signal TTTT or (T) ₄		Termination signal TTTTT or (T) ₅		Termination signal TTTTTT or (T) ₆		Termination signal TTTTTT or (T) ₇	
	Average of 3'-trailer length ± S.D. (nt)	Number of tRNAs analysed (3'-trailer lengths with >20 nt excluded)	Average of 3'-trailer length ± S.D. (nt)	Number of tRNAs analysed (3'-trailer lengths with >20 nt excluded)	Average of 3'-trailer length ± S.D. (nt)	Number of tRNAs analysed (3'-trailer lengths with >20 nt excluded)	Average of 3'-trailer length ± S.D. (nt)	Number of tRNAs analysed (3'-trailer lengths with >20 nt excluded)
H. sapiens	8.1 ± 3.7	412	8.4 ± 3.6	140	8.2 ± 3.9	65	7.4 ± 3.7	45
M. musculus	7.9 ± 3.6	343	7.9 ± 3.5	141	7.9 ± 3.4	55	7.6 ± 3.0	39
D. melanogaster	8.5 ± 4.0	247	8.2 ± 3.9	214	8.1 ± 3.9	162	7.1 ± 3.4	79
A. thaliana	4.6 ± 4.3	532	4.7 ± 4.3	441	5.2 ± 4.4	251	4.9 ± 4.1	133
S. cerevisiae	3.0 ± 2.6	269	3.5 ± 3.0	266	3.6 ± 3.2	214	3.9 ± 3.6	132
S. pombe	3.9 ± 3.6	162	4.3 ± 3.8	154	3.9 ± 3.7	86	4.2 ± 4.5	36
T. thermophila	1.5 ± 1.5	686	1.7 ± 1.9	674	2.2 ± 2.9	545	2.5 ± 3.3	312

Thus, our interpretation should be valid irrespective of the actual *T. thermophila* RNA polymerase III terminator T-length requirement. We've added a sentence to the manuscript (page 16) and a supplementary table (**Supplementary Table S2**) and updated the data in

Additional **Supplementary Data 5** (columns R, S, T) to reflect this and thank the reviewer for the suggestion.

5. Fig S7A-B – should be presented as a boxplot including statistics

Since the n of the relevant TGIRT sequencing was only n=3, we feel a boxplot would not be very representative of the data.

Error bars for the data in **Supplementary Figure 7A-B** are provided for equivalent data in **Supplementary Figure S2**. We have redrawn **Supplementary Figure 7A-B** with error bars below:

We have replaced **Supplementary Figure S7A-B** with the version containing error bars. We feel it is important to note that we have drawn no significant conclusion from these data in the manuscript (page 16): “suggesting that nascent pre-tRNAs containing a longer 3’-trailer sequence may have a slightly lower binding affinity for Mlp1”.

Reviewer #2 (Remarks to the Author):

In the presented manuscript, the authors functionally characterize the Tetrahymena thermophila protein Mlp1, that has previously been classified as a genuine La protein. Members of this protein family bind to the UUU-3’OH containing pre-tRNAs and coordinate the process of 5’-leader and 3’-trailer removal by specific RNAses (RNase P and RNase Z). In addition, the presence of La proteins is known to support pre-tRNA folding and maturation. As the ThMlp1 sequence does not display an obvious RRM, it was not clear, if Mlp1 acts as a genuine La protein and if the mechanism would be similar to its known homologues.

The authors perform comprehensive analyses, using a variety of experimental approaches. The

study is well executed and the scientific claims are fully supported by the presented results. The used methodology is well documented and I could not detect any obvious signs of misinterpretations and/or overstatements. The manuscript adequately describes and discusses the available literature in the field. In detail, the authors show that Mlp1 indeed functions as La protein, binds pre-tRNAs, regulates pre-tRNA processing and supports tRNA folding. Furthermore, the authors discovered unique mechanistic features, which can be associated with the presence of unusually short 3'-trailers in Tetrahymena – a characteristic that the authors also describe for the first time. Therefore, the study indeed presents an appealing and novel mechanisms of pre-tRNA processing that is coordinated by Mlp1. The work might provide a reference study that allows to discriminate between canonical and non-canonical routes of pre-tRNA processing in the future. Hence, I in principle support the publication in Nature Communication, but I would suggest to use state-of-the-art structure prediction approaches to present more complete structural models of full length Mlp1.

We were grateful that the reviewer viewed our work favourably and thank them for their comments.

Major issue

I would suggest to use alphafold2 (or similar algorithms) to predict the structure of Mlp1 de novo. The presented homology model is obviously already very useful to identify crucial residues in Mlp1, but considering the low sequence conservation it is not surprising that it falls short in providing information about other regions of the protein. Considering the presented biochemical results, I could imagine that a cryptic RRM domain could actually be present in ThMlp1. In addition, it would be useful for the reader to have at least a tentative model of the DUF3223 domain, which also could present RNA recognition surfaces as it seems to be important for the described RNA chaperoning function as well.

We thank the reviewer for this comment, and attach here in this response to reviewers an alphafold prediction we have performed for Mlp1. As expected, alphafold predicted a La motif similar to those that have been described previously, and a fold for the DUF3223 domain very similar to that which has been demonstrated from an NMR structure for a related domain from *Rhodospirillum rubrum*. Critically, alphafold indeed predicted the lack of an RRM domain in the expected location, relative to the La motif. In human La, the RRM is approximately 10 amino acids C-terminal to the La motif, which is then followed by a second variant xRRM domain, another ~ 25 amino acids C-terminal from RRM1. In Mlp1, the RRM1 is indeed lacking, and there is an xRRM (distinguished from the classic RRM by a clear alpha helix obscuring the canonical RNA binding surface) approximately 25 amino acids C-terminal to the La motif, with these intervening amino acids forming an extended alpha helix. Relative to human La, it appears that the RRM1 has been skipped, with the La motif going directly into the xRRM after an extended alpha helix. Fascinatingly, the DUF3223 then comes back to be proximal to the La motif after another extended alpha helix (which extends from the alpha-3 helix of the xRRM), placing the DUF3223 region closer to the space that would have been expected to be occupied by RRM1. Thus, alphafold agrees that the expected RRM1 domain is lacking.

hLa architecture (RRM2 α = xRRM; Maraia et al., 2017)

[redacted]

Mlp1 alphafold prediction:

Human La module (LaM + RRM1):

We are currently attempting to perform structural studies on the region where the RRM1 domain would be expected to be located. We have successfully purified large quantities of the (13C and 15N labelled) 1-95 and 95-225 regions (the La motif and the region between the La motif and the DUF3223) and have acquired excellent NMR data for structural characterization, in collaboration with Maria (Sasi) Conte's group at King's College, London, who is an expert on solving the structure of La and La-related proteins. Our concern is that by including an AlphaFold prediction (which we have performed) in the current manuscript, this will provide readers a false sense that the structure of the 95-225 region is "finished". This is especially true seeing as the AlphaFold simulation we performed had a number of low confidence regions,

especially in the borders between predicted globular domains, thus further increasing the risk of what may be taken by the reader to be a “finished” structure is in fact inaccurate.

The current manuscript is grounded in the principle that the 95-225 domain lacks an RRM in the classic arrangement expected for a La module, and the AlphaFold prediction we performed is certainly consistent with that. Seeing as the AlphaFold prediction does not contradict or jeopardize the narrative in our current work, and since we did not intend to write a paper that attempts to speculate on structure between 95-225 (only that it does not have the classic LaM-RRM arrangement), we are concerned that to include this prediction in the current work will only serve to undermine our future work, which will include rigorous structure-function analysis combined with new biochemical experiments grounded in the new insights acquired from that structure.

We feel that our claim that Mlp1 does not contain the classic LaM-RRM1 arrangement is sufficiently supported by the current data, and our preference is to not include an AlphaFold prediction in this paper. Our revised manuscript does not currently include this. We are hopeful that, knowing that we are attempting to solve the actual structure of Mlp1 95-225, the reviewer would agree that inclusion of the AlphaFold prediction at this time would be counterproductive. However, we appreciate the relevance of the point raised, and if the reviewer still prefers the inclusion of the prediction, we will include it.

Minor issues

- *Please introduce Mlp1 in the first paragraph of the results section – otherwise this paragraph reads very confusing*

We added the sentence “A recent phylogenetic study in eukaryotes of all kingdoms identified Mlp1 as a new atypical genuine La protein in alveolates, containing a highly conserved LaM without an adjacent RRM (Deragon, 2020)” to the beginning of the result section.

- *Please remove the redundant “nM” labels for the truncated constructs in Table 1*

This has been completed.

Reviewer #3 (Remarks to the Author):

In this manuscript, Kerkhofs et al study the mechanism of Mlp1, a La-like protein in T. thermophila that seems to confer to the species an alternative La protein dependency in tRNA maturation as compared with other eukaryotes. They conclude that: 1) Mlp1 is likely a La protein, 2) Mlp1 has lower affinity for pre-tRNAs and their 3' trailer sequences than human La protein, 3) Mlp1 has a tRNA chaperone function, and 4) Mlp1, unlike other RNA recognition motif-1 (RRM1)-containing La proteins, destabilizes tRNA 3' trailer sequences. This work provides a useful characterization of an otherwise poorly understood mechanism in T. thermophila and provides evidence for proper interpretation of tRNA maturation in species

lacking genuine characterized La proteins. Nonetheless, I have some concerns regarding data and conclusions presented in the manuscript, as outlined below:

We are pleased that the reviewer found our characterization useful and thank them.

1. In Figure 2A/B, the authors demonstrate that both Mlp1 and hLa have a binding preference for pre-tRNAs over mature tRNAs. While the data is compelling, the authors should comment on why there is increased variability in the enrichment of pre-tRNAs when looking across isoacceptor classes. It seems that hLa has high preference for all pre-tRNAs except for Cys tRNAs, for which there is no preference for pre- over mature species. There are several isoacceptor classes for which Mlp1 does not exhibit preference for the pre-tRNA. Could it be inherent to the trailers of these classes? It seems that the preference correlates with the amino acids in an alphanumeric manner, and is this some sort of technical artifact in the data?

This is indeed an interesting observation for which we don't have a clear answer, except to concur that human La also showed varying degrees of enrichment for a number of classes of pre-tRNA (including but not limited to Cys; Gogagos et al., 2017) for reasons that were not clear. We have double checked and confirmed that the pattern is not due to an artifact related to the alphanumeric order. This breadth could be due to a yet uncharacterized La binding determinant on certain tRNAs. Alternatively, it could be due to a less interesting issue relating to the workflow, such as quantitation bias relating to tRNA modifications on certain tRNAs that are refractory to cDNA synthesis, similar to how the location of uracils on certain pre-tRNAs relative to hLa contacts could have introduced bias into the PAR-CLIP reads in Gogagos et al., 2017. We attempted to find a correlation between anticipated trailer length and Mlp1 affinity (see **Supplementary Figure S8a,b**) and found a small but not very convincing correlation (although it was a better correlation than what we found from the equivalent data for hLa), which we noted. Overall, however, we feel the data are consistent with Mlp1 enriching pre-tRNAs over mature tRNAs generally, even if the enrichment is not as strong enrichment as with hLa, which is consistent with our other data and other language we have used in the manuscript. For example, we have speculated (Discussion) that this lesser discrimination may be related to the lack of the RRM (*"The impaired ability of Mlp1 to discriminate the U₋₂ residue, as well as uridylates more generally, suggest that the lack of the RRM1 results in a relatively lower ability of Mlp1 to differentiate UUU-3'OH containing RNAs"*).

2. The experimental design for results presented in Figure 3A/B could be clarified. Are human or tetrahymena mature and pre-tRNAs used for the competition assay? Are these a mixture of all tRNA species or enriched for particular isodecoders?

Having extensively used *T. th* tRNA-Leu^{AAG} in our EMSA experiments in **Figure 2**, we used the same pre- and mature tRNA in the competition experiments in **Figure 3**. This is now clarified in the **Figure 3 (B&C)** figure legend, as well as **Figure 2C**.

3. It is unclear why the U>C change is made in the 4U construct used to generate Figure 3D/E. Why not change U to A or G, and would this be expected to impact the results shown here? The

authors should demonstrate that there is no nucleotide preference for the phenomenon described when comparing hLa and Mlp1 data.

We used uridylate to cytidylate substitutions in the competition assays because it was these U to C substitutions that had the greatest effect in the equivalent competition assays for human La (Teplova et al., 2006), which we also repeated in this work so as to be able to have a side-by-side, “apples to apples” comparison. This is now noted in the revised manuscript.: “...we performed competition EMSAs for hLa and Mlp1 using the radioactively labeled UUU-3’OH containing RNA CUGCUGUUUU-3’OH RNA (hence referred to as wild type 4U) and unlabelled RNAs carrying specific U to C variations to the terminal uridylates in this sequence, as previous work indicated the U to C substitution had the greatest effect on RNA substrate discrimination by human La¹⁸.” We have also added new EMSA data confirming that, similar to human La, Mlp1 has no measurable affinity for a C10 homopolymer (new **Supplementary Figure S5b**).

4. The illustrations adjacent to Figure 4B/C are, in certain cases, difficult to map to the bands of the northern blots. Perhaps some sort of boxing around the bands could be done to improve this. That said, the identities of all of the bands on these blots is not always clear with the provided illustrations, especially in Figure 4C. There are bands for which there are no corresponding illustrations, leading to difficulty in independent interpretation of the blots. The authors should fix this to provide comprehensive annotation where possible. Furthermore, it is unclear how the authors mapped each of the illustrations to their corresponding bands. The authors should provide additional evidence by means of 3’ and 5’ trailer probes and/or 5’ and 3’ processing defective strains to confirm the identities of their northern blot bands.

The advantage of using these two particular pre-tRNAs in our assessment of pre-tRNA species is that the identity of each band has been rigorously established in preceding work, specifically in Intine et al., 2000, Intine et al., 2002 (both *Mol Cell*) and subsequent papers. We have added new annotations (see below) so as to make the assignment of the relevant bands clearer.

5. It seems to be the case, with the northern blot in Figure 4C, that Mlp1 stabilizes tRNAs with a fully intact 3’ trailer sequence, as indicated by the adjacent illustrations. The authors should discuss this inconsistency in light of the conclusions made in this paper that limiting Mlp1 expression stabilizes 3’ trailers.

We believe the Mlp1 stabilizes a trailer trimmed or nibbled intermediate relative to Sla1p, specifically the band that is labelled d in the leader probed blot Figure 4C, which corresponds to the top-most band in the trailer probed sequence below that. Our new annotation of bands in the revised figures should also be helpful in this distinction. The increased reactivity of this band with the 5’-leader probe relative to the 3’-trailer probe is consistent with this, as is its diminished reactivity using the trailer probe relative to the trailer stabilized bands in the Sla1p lane. Unfortunately, the trailer probe still has some overlap with the sequence found at the 3’

end of what will become the mature tRNA (leading up to the discriminator base), as the relevant 3' trailers are not long enough to bind to the trailer probe exclusively, without any overlap to the tRNA body. Thus for a 3' nibbled trailer, we would expect a slightly smaller species with lower (but not zero) signal when binding the trailer probe, which is exactly what we see. We performed the 3'RACE in order to add resolution to this result, and that experiment was further consistent with this. We hope the added annotation of bands to the blots further clarifies this.

6. The authors should discuss the presence of mature tRNA sequence that seems to have been processed in Figure 4D. What is the nature of this loss of sequence in the s-bound Lys and Tyr tRNAs?

We addressed this comment with Reviewer 1: we believe the diminished discrimination of Mlp1 relative to hLa leads to Mlp1 binding to CCA added but intron containing pre-tRNA species, prior to their export to the cytoplasm for splicing. We have added text on page 13/14 to address this point.

Minor edits:

1. 'suppressor tRNA stabilization' is indicated on Figure 4C. I think that this should not be indicated as a suppressor tRNA.

We have added language to the relevant figure legend ("with suppressor tRNA stabilization levels (Figure 4a) shown above the blot") so as to clarify the purpose of this, which is to be able to align pre-tRNA maturation of pre-tRNA-LysCUU with the related results from the tRNA mediated suppression assay.

2. It is somewhat difficult to evaluate the effectiveness of suppression between some of the colonies in Figure 4A due to differing background color tone. The difference between + and ++ is not clear to me.

The figure has been updated to be able to pay closer attention to the background colour. We have added wording to the Figure 4A figure legend in order to make the rationale for how colour assessment is made clearer. The full image can be found in the Source Data file.

REVIEWERS' COMMENTS

Reviewer #1 (Remarks to the Author):

Reviewed manuscript sounds good and is easier to follow. The authors provided links between sections that improved the flow of the MS, and addressed other points raised. I support publication of the manuscript in the present form.

Reviewer #2 (Remarks to the Author):

The authors have addressed and answered the raised issues. The additional alphafold prediction indeed supports the initial finding and should be included in the current manuscript. I disagree with the author's statement " by including an AlphaFold prediction (which we have performed) in the current manuscript, this will provide readers a false sense that the structure of the 95-225 region is "finished". After all, an experimentally determined structures also do not "finish" the analyses. I assume any future structural analysis of Mlp1 will anyway rather focus on the complex with pre-tRNAs. The structural analysis of individual protein domains will not provide sufficient insights to make any far reaching conclusion that go beyond the presented structural predictions. Therefore, I would kindly ask the authors to include the analyses in the manuscript. In my opinion, the manuscript is suitable to be published in Nature Communications only after adding the AF analysis to the manuscript.

Reviewer #3 (Remarks to the Author):

The authors have satisfied my concerns.

REVIEWER COMMENTS

Reviewer #1 (Remarks to the Author):

Reviewed manuscript sounds good and is easier to follow. The authors provided links between sections that improved the flow of the MS, and addressed other points raised. I support publication of the manuscript in the present form.

We thank the reviewer for their contribution to improving our manuscript.

Reviewer #2 (Remarks to the Author):

The authors have addressed and answered the raised issues. The additional alphafold prediction indeed supports the initial finding and should be included in the current manuscript. I disagree with the author's statement "by including an AlphaFold prediction (which we have performed) in the current manuscript, this will provide readers a false sense that the structure of the 95-225 region is "finished". After all, an experimentally determined structures also do not "finish" the analyses. I assume any future structural analysis of Mlp1 will anyway rather focus on the complex with pre-tRNAs. The structural analysis of individual protein domains will not provide sufficient insights to make any far reaching conclusion that go beyond the presented structural predictions. Therefore, I would kindly ask the authors to include the analyses in the manuscript. In my opinion, the manuscript is suitable to be published in Nature Communications only after adding the AF analysis to the manuscript.

In agreement with the editor, the alphafold prediction will be included in the manuscript in the context of the publicly available peer review file. The prediction was included in the previous response to reviewers. We thank the reviewer for their contribution to improving our manuscript.

Reviewer #3 (Remarks to the Author):

The authors have satisfied my concerns.

We thank the reviewer for their contribution to improving our manuscript.